# RetrievalFormer: A Dual-Encoder Transformer for Efficient Approximate Nearest Neighbor Retrieval and Cold-Item Recommendation

## Abstract

We propose RetrievalFormer, a transformer-based dual-encoder recommender architecture that combines competitive accuracy with strong transformer-based sequential baselines, efficient Approximate Nearest Neighbor (ANN) retrieval, and the ability to score feature-described items that are unseen during training. Our architecture uses an attention-based heterogeneous feature encoder that aggregates item and user attributes via shared embedding tables and an AttentionFusion module, so that the resulting user and item representations lie in a shared embedding space suitable for ANN search. On Amazon and MovieLens benchmarks, RetrievalFormer achieves competitive recommendation accuracy, reaching 86–91% of the Recall@20 of strong transformer-based sequential baselines while enabling up to $288\times$ lower latency at a 10M-item scale via ANN retrieval. On MovieLens-1M, RetrievalFormer attains Recall@20 of 0.337. In cold-start experiments where entire items and all of their interactions are held out during training, RetrievalFormer successfully recommends completely unseen items from their features in a leave-one-out cold (LOOC) protocol with zero item leakage between training and evaluation, in which ID-softmax transformer baselines cannot produce scores for such items at all, and it outperforms a strong content-based baseline on a 100% cold-start production dataset. Our approach enables practical deployment of efficient recommendations at scale, offering a compelling trade-off between model accuracy and serving efficiency.

## 1 Introduction

Transformer-based sequential recommenders (e.g., SASRec, BERT4Rec) have achieved state-of-the-art accuracy in next-item prediction by leveraging self-attention over user behavior sequences (Kang & McAuley, 2018; Sun et al., 2019). These models treat recommendation as a classification over all items in the catalog: given a sequence of past items, the transformer produces a probability distribution over the entire item vocabulary for the next item (Vaswani et al., 2017). While effective, this approach has two key shortcomings in real-world settings.

This ID-softmax formulation simultaneously causes two related issues. First, the output layer must compute scores for all $N$ items in the catalog, incurring an $O(Nd)$ cost per prediction that dominates the $O(L^2d)$ cost of self-attention once $N$ is large. Second, because the output layer contains one parameter vector per training-time item ID, items whose IDs never appear in training cannot be scored at all, even if rich item features are available at inference time.

First, scoring all items via a full softmax is computationally expensive for large catalogs. Serving such models in production requires scanning through millions of item embeddings for each prediction, leading to high latency and resource costs (Su et al., 2023). For example, Kersbergen et al. (2024) report that a transformer model with a 20-million item catalog required multiple high-end GPU servers to meet a 50ms p90 latency, incurring thousands of dollars per month in deployment cost (Kersbergen et al., 2024).

Second, classical transformers struggle with item cold-start: new items cannot be effectively recommended until the model is retrained or updated (Volkovs et al., 2017; Schein et al., 2002). In dynamic domains with rapid item churn (e.g., news or ephemeral marketing content), the delay in recommending new items is problematic.

We propose RetrievalFormer, a dual-encoder sequential recommender that directly addresses both of these issues by reframing next-item recommendation as a retrieval problem. A transformer-based user tower encodes the interaction history into a user embedding, and a feature-based item tower encodes items from their attributes into item embeddings, with both towers trained jointly so that recommendations are produced by dot-product similarity in a shared embedding space rather than a softmax over item IDs. By decoupling user and item representations, our model enables efficient retrieval: at serving time, a user's latest interaction sequence is encoded into a query embedding, and the top-$K$ candidate items are retrieved by Approximate Nearest Neighbor (ANN) search in the item embedding space, instead of computing a full softmax over all items. Concretely, the relevance score between a user $u$ and an item $i$ is

$$s(u, i) = f_u(\text{history}_u)^\top f_i(\text{features}_i),$$

and at serving time we retrieve the top-$K$ items for a given user by performing ANN search over the pre-computed item embeddings $f_i(\cdot)$. This retrieval paradigm leverages highly optimized ANN indexes (e.g., HNSW graphs or vector quantization) to find top candidates in sub-linear time (Johnson et al., 2019; Malkov & Yashunin, 2018), circumventing the costly softmax over the entire catalog. In essence, RetrievalFormer achieves transformer-like recommendation quality while operating at the speed of ANN retrieval.

Moreover, the item tower directly computes representations from item content and attributes, so new items can be recommended zero-shot, addressing the item cold-start problem without any retraining or extension of the model's vocabulary.

Our approach also introduces an attention-based heterogeneous feature encoder to enrich both user and item representations. Modern recommender data is heterogeneous, with information such as item text descriptions, categories, images, and contextual tags, as well as user profile features. Rather than using only IDs or simple feature concatenation, we apply a self-attention fusion mechanism to each set of features describing an entity (an item or an interaction). This allows the model to learn complex interactions between different feature modalities in a data-driven way. For example, an item's textual description and its category label can attend to each other to produce a more informative item embedding. This design draws inspiration from Set Transformer architectures (Lee et al., 2019) and feature interaction learning (Song et al., 2019), enabling permutation-invariant aggregation of arbitrary feature sets. Importantly, this attention fusion mechanism is used throughout our architecture, in the item tower for combining item metadata, in the user interaction history for fusing features of historical items, and in the user tower for processing the resulting token sequence. We further share embedding lookup tables for features across the user and item towers, so that a feature (e.g., a brand ID or a word embedding) has a consistent representation regardless of where it is used. This weight sharing improves training efficiency and alignment between the two towers, as the model can leverage the same semantic signal in multiple contexts.

We validate RetrievalFormer on standard benchmarks, finding competitive accuracy versus transformers while achieving 288× speedup at 10M items. Our contributions: (1) a two-tower architecture achieving competitive accuracy with efficient ANN retrieval, (2) attention fusion for heterogeneous features outperforming simple pooling, (3) zero-shot cold-start capability through feature-based encoding, and (4) rigorous evaluation demonstrating practical trade-offs between accuracy and efficiency.

## 2 RELATED WORK

**Sequential Recommendation and Transformers.** Sequential recommenders model the dynamic sequence of user-item interactions to predict a user's next interest. Early approaches used Markov Chains or RNNs (Hidasi et al., 2015; Li et al., 2017; Tang & Wang, 2018; Wu et al., 2017), but recent advances are dominated by self-attention mechanisms. SASRec (Kang & McAuley, 2018) introduced the use of unidirectional Transformer encoder layers to capture which previous items in the sequence are relevant for predicting the next one. Variants like BERT4Rec extended this with

bidirectional transformers and a Cloze task for training (Sun et al., 2019). These models learn item embeddings and position embeddings, and use multi-head attention to capture long-range dependencies in user behavior. While very effective in accuracy, a core limitation is that they produce predictions by a softmax over the entire item vocabulary at each time step (Kang & McAuley, 2018; Sun et al., 2019). This does not scale well to large catalogs due to the computational cost and memory footprint of the output layer. Recent work has noted the inference bottleneck of such models: even with optimizations, serving a transformer sequential model for millions of items can be prohibitively slow or costly (Su et al., 2023).

**Two-Stage and Retrieval Models in Recommenders.** In industry-scale recommender systems, a common solution is a two-stage pipeline: first retrieve a set of candidates, then apply a more precise ranking model (Covington et al., 2016). The candidate retrieval stage often uses lightweight models (e.g., matrix factorization (Krichene & Rendle, 2020) or two-tower neural networks (Yi et al., 2019)) that can handle a very large item pool efficiently (Yi et al., 2019; Huang et al., 2020a; Eksombatchai et al., 2018; Grbovic & Cheng, 2018). Our work follows this paradigm in spirit: RetrievalFormer's user and item towers correspond to a learned retrieval model producing candidate item embeddings. The key difference is that we aim to approach the accuracy of strong transformer-based sequential models in the retrieval stage itself, rather than using a simplistic retriever, effectively collapsing the quality of a powerful sequential model into an ANN-friendly form.

**Attribute-Enriched and Cold-Start Recommendation.** Another line of related work is utilizing content features and attributes to improve recommendations, especially under sparse data or cold-start scenarios (Schein et al., 2002; Zhou et al., 2022; de Souza Pereira Moreira et al., 2021; Pancha et al., 2022). Many recommender models have been extended to incorporate side information such as item descriptions, knowledge graph entities, or user profile data. For sequential recsys, recent methods like AttrFormer explicitly model item attributes alongside IDs in the attention mechanism (Liu et al., 2025). AttrFormer augments an ID-softmax transformer with attribute-aware attention and achieves strong accuracy on Amazon benchmarks, but it still predicts over a fixed item vocabulary and cannot score items whose IDs never appear during training. In contrast, RetrievalFormer decouples item representations into a feature-based item tower within a dual-encoder design, enabling direct scoring of unseen items from their attributes and making the model naturally compatible with ANN retrieval.

**Approximate Nearest Neighbors for Recommendation.** Fast ANN search has seen rapid progress, with algorithms like IVF, HNSW, and PQ enabling vector search on billions of points within milliseconds (Johnson et al., 2019; Malkov & Yashunin, 2018). Our contribution ensures using ANN does not sacrifice recommendation quality, by training to produce a discriminative embedding space, we achieve both high accuracy and low latency. We do not propose a new ANN algorithm in this work; rather, we design a sequential recommender whose dual-encoder architecture and training objective produce an embedding space that is well aligned with standard ANN indexes such as IVF-PQ and HNSW, so that they can be used for serving without sacrificing recommendation quality.

A large body of work aims to reduce the computational cost of neural recommendation models through techniques such as sampled or approximate softmax, model compression and distillation, and explicit two-stage candidate-generation plus re-ranking pipelines. These approaches reduce the effective number of items scored or the size of the re-ranking model, but they typically retain an ID-softmax formulation or a separate heavy re-ranker. By contrast, RetrievalFormer reformulates sequential recommendation itself as a dual-encoder retrieval problem so that efficient ANN search over item embeddings becomes the native serving mechanism.

## 3 METHODOLOGY

We present RetrievalFormer, a dual-encoder architecture that achieves competitive sequential recommendation accuracy compared to strong transformer-based baselines while enabling efficient ANN-based retrieval. Our approach addresses the fundamental scalability limitation of transformer recommenders, the $O(N)$ inference cost of scoring all items, by decoupling item representations from user sequence modeling. This section describes our architecture design, the attention fusion mechanism for heterogeneous features, and the training methodology that enables both accuracy and efficiency.

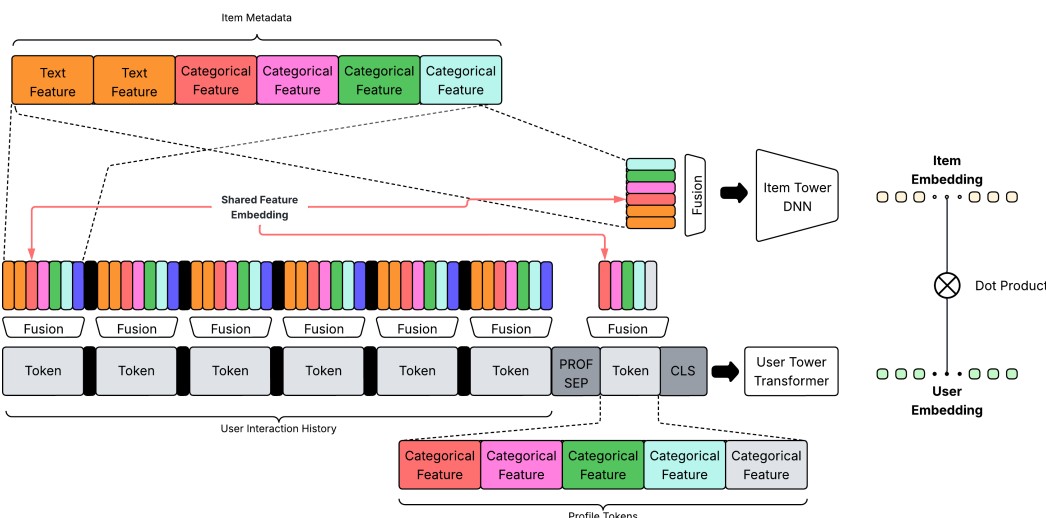

Figure 1: Raw user and item features are embedded and fused by AttentionFusion, then passed through the user and item towers to produce embeddings in a shared space. At serving time, user embeddings query an ANN index over pre-computed item embeddings.

## 3.1 OVERALL ARCHITECTURE

RetrievalFormer employs a dual-encoder design with asymmetric towers optimized for their respective roles (Figure 1). The item tower $f_i(\cdot)$ encodes each item's heterogeneous features into a dense embedding $\mathbf{y} \in \mathbb{R}^d$ that can be pre-computed and indexed. The user tower $f_u(\cdot)$, implemented as a transformer, processes the user's interaction sequence to produce a query embedding $\mathbf{x} \in \mathbb{R}^d$. At serving time, recommendations are generated through approximate nearest neighbor search for items $\mathbf{y}$ that maximize $\mathbf{x}^\top \mathbf{y}$, avoiding the computational bottleneck of exhaustive scoring.

The key insight is that by learning a shared embedding space through contrastive training, we can leverage the same representations for both training (via InfoNCE loss) and inference (via ANN retrieval). This design choice changes the effective scaling from $O(N)$ to empirically sub-linear growth in practice when using standard ANN indexes, while maintaining recommendation quality, as we demonstrate in Section 4.

Formally, let $x_u = f_u(\mathbf{h}_u)$ denote the user embedding produced by the user tower from the interaction history $\mathbf{h}_u$, and let $y_i = f_i(\mathbf{z}_i)$ denote the item embedding produced by the item tower from the item features $\mathbf{z}_i$. The relevance score is

$$s(u, i) = x_u^\top y_i,$$

and the top-$K$ recommendations for user $u$ are the $K$ items with the largest values of $x_u^\top y_i$, found via ANN search over $\{y_i\}$.

## 3.2 ATTENTION FUSION FOR HETEROGENEOUS FEATURES

Modern recommendation systems must handle diverse feature types: text descriptions, categorical attributes, numerical values, and interaction signals. We introduce an attention-based fusion mechanism that learns to dynamically weight and combine these heterogeneous features, moving beyond simple concatenation or averaging approaches.

### 3.2.1 FEATURE FUSION MECHANISM

Given features $\mathcal{F} = \{f_1, ..., f_M\}$ describing an entity, we embed each feature using shared lookup tables and project to a common dimension:

$$\mathbf{H} = [\mathbf{W}_1 \mathbf{E}_{f_1}(f_1); ...; \mathbf{W}_M \mathbf{E}_{f_M}(f_M)] \in \mathbb{R}^{M \times d} \tag{1}$$

In our experiments, $\mathcal{F}$ includes single-valued categorical features (e.g., item category, brand), multi-valued categorical features (e.g., tags), and text-derived features (e.g., token IDs from titles or descriptions). Single-valued categorical features are encoded as a single embedding vector per feature, while multi-valued features are encoded by aggregating the embeddings of all values for that feature (either via mean pooling or attention). Text features are treated as multi-valued categorical features over a token vocabulary. AttentionFusion applies multi-head self-attention over the set of feature embeddings for an item or user, followed by pooling, to produce a single fixed-dimensional representation.

We apply multi-head self-attention (Vaswani et al., 2017) with residual connections and layer normalization:

$$\mathbf{Z} = \text{LayerNorm}(\mathbf{H} + \text{MultiHeadAttn}(\mathbf{H}, \mathbf{H}, \mathbf{H})) \tag{2}$$

$$\mathbf{U} = \text{LayerNorm}(\mathbf{Z} + \text{FFN}(\mathbf{Z})) \tag{3}$$

$$\mathbf{z} = \text{MeanPool}(\mathbf{U}) \in \mathbb{R}^d \tag{4}$$

This mechanism is permutation-invariant and handles variable-length feature sets, learning complex feature interactions through attention weights. The same fusion architecture is applied consistently at three levels: (1) item metadata fusion, (2) interaction context fusion, and (3) user profile fusion.

### 3.2.2 SHARED EMBEDDING DESIGN

A critical design choice is sharing embedding tables across towers. When a categorical feature (e.g., "electronics") appears in different contexts, as an item category, user preference, or interaction attribute, it uses the same embedding vector. This parameter sharing reduces parameters by approximately 3× in our implementation, enables knowledge transfer between representations, improves cold-start generalization, and ensures consistent feature semantics.

Sharing embedding tables across user profile, item metadata, and interaction history for the same feature types not only reduces the number of parameters, but also encourages a consistent semantic space for these features. This is particularly important for cold-item generalization, since it allows the model to interpret the same attribute (e.g., a brand or category) consistently whether it appears in a user profile, a historical interaction, or a newly introduced item.

### 3.3 ITEM TOWER: FEATURE-BASED ENCODING

The item tower computes dense embeddings from item features. For item $i$ with features $\mathcal{F}_i = \{f_1^{(i)}, ..., f_M^{(i)}\}$:

$$\mathbf{y}_i = \text{AttentionFusion}(\mathcal{F}_i) \in \mathbb{R}^d \tag{5}$$

This feature-based design enables zero-shot generalization where new items receive embeddings immediately from their features. The tower leverages shared feature embeddings and fusion weights, providing scalability and robustness through graceful handling of missing features via attention masking.

Because the item tower depends only on item-side features and not on user history, we can precompute $y_i = f_i(\mathbf{z}_i)$ for all items offline and build an ANN index over these embeddings, so that online serving only needs to compute the user embedding $x_u$ and perform an ANN query.

### 3.4 USER TOWER: TRANSFORMER OVER ENRICHED SEQUENCES

The user tower processes interaction sequences through a transformer encoder, but critically, each sequence element is an enriched representation combining item features with interaction context.

### 3.4.1 INTERACTION REPRESENTATION

For each historical interaction $e_t$ involving item $i_t$, we create enriched tokens through two-stage fusion:

$$\mathbf{h}_{i_t} = \text{AttentionFusion}(\text{ItemFeatures}(i_t)) \tag{6}$$

$$\mathbf{z}_t = \text{AttentionFusion}(\mathbf{h}_{i_t} \oplus \text{InteractionContext}(e_t)) \tag{7}$$

where $\oplus$ denotes feature concatenation and InteractionContext includes interaction type (click, purchase), explicit feedback (ratings), and contextual signals (device, timestamp). This two-stage process captures not just *what* items were interacted with, but *how* and *when*.

In all of our experiments, we use the same number of transformer layers and hidden dimension as in the corresponding transformer baselines (e.g., SASRec and BERT4Rec) so that differences in accuracy are attributable to the dual-encoder formulation rather than model capacity; detailed hyperparameters are provided in Section 4.1 and Appendix J.

### 3.4.2 SEQUENCE CONSTRUCTION

The transformer processes the sequence:

$$\mathbf{S} = [\mathbf{z}_1, ..., \mathbf{z}_T, [\text{SEP}], \mathbf{p}_u, [\text{CLS}]] \tag{8}$$

where $\mathbf{p}_u = \text{AttentionFusion}(\text{UserFeatures})$ encodes static user attributes. The final [CLS] token representation, after transformer processing with causal masking, becomes the user embedding $\mathbf{x}_u$.

This design enables the transformer to model sequential patterns over semantically rich tokens, improving both accuracy and generalization.

### 3.5 TRAINING METHODOLOGY

We train RetrievalFormer using contrastive learning to learn a shared embedding space where users and their next items are close while being far from non-relevant items. For a batch of $B$ user-item pairs with embeddings $\{(\mathbf{x}_i, \mathbf{y}_i)\}_{i=1}^{B}$, we optimize the InfoNCE loss (Oord et al., 2018):

$$\mathcal{L}_{\text{InfoNCE}} = -\frac{1}{B} \sum_{i=1}^{B} \log \frac{\exp(\mathbf{x}_i^\top \mathbf{y}_i / \tau)}{\sum_{j=1}^{B} \exp(\mathbf{x}_i^\top \mathbf{y}_j / \tau)} \tag{9}$$

where $\tau$ is a temperature hyperparameter. This objective treats all other items in the batch as negatives, efficiently approximating the full softmax over the catalog.

To address popularity bias and improve coverage of tail items, we employ Mixed Negative Sampling (MNS) (Yang et al., 2020), augmenting each batch with uniformly sampled items from the catalog. This ensures diverse negative signals across the entire item distribution, preventing the model from over-optimizing on popular items while neglecting rare ones.

The combination of InfoNCE and MNS is particularly important for RetrievalFormer's training. The contrastive objective encourages both alignment between user and positive item embeddings and uniformity of the overall embedding distribution on the hypersphere (Wang & Isola, 2020), which helps to avoid representation collapse and makes the learned space more suitable for ANN retrieval. A brief discussion of these alignment and uniformity properties, together with implementation considerations for mixed negative sampling, is provided in Appendix C.

## 4 EXPERIMENTS

We structure our experimental evaluation around four research questions: **RQ1** examines whether RetrievalFormer can achieve competitive recommendation accuracy compared to state-of-the-art transformer-based sequential models on standard benchmarks. **RQ2** investigates how the heterogeneous feature inputs and architectural choices (attention fusion, shared embeddings, context tokens) contribute to the model's performance. **RQ3** evaluates how well RetrievalFormer handles unseen items and whether it can effectively recommend items that were never in the training set. **RQ4** measures the inference efficiency of RetrievalFormer using ANN search and compares it to a conventional Transformer model that scores all items.

## 4.1 Experimental Setup

**Datasets:** We evaluate on three public datasets used in prior Transformer recommender research: Amazon Beauty, Amazon Toys & Games, and MovieLens-1M. For Amazon, we use the sequential rating/review data from McAuley et al. (2015), focusing on users with at least 5 interactions. For MovieLens-1M, we use the 1 million movie ratings, treating a rating as an implicit interaction. For RQ1, RQ2, and RQ4, we use the same data splits, features, and preprocessing as Liu et al. (2025) for direct comparability, where a standard leave-one-out (LOO) approach holds out the last item for testing, the second-to-last for validation, and the remainder for training. For RQ3 (cold-start evaluation), we use a separate Leave-One-Out Cold (LOOC) protocol that ensures test items are completely absent from training; see Section 4.4 and Appendix F for details. We also use a proprietary Email Campaign dataset as a case study for extreme item cold-start (each "item" is a marketing email, and new campaigns launch daily with no historical interactions).

**Baselines:** For RQ1, we compare RetrievalFormer to representative sequential recommenders: SASRec (Kang & McAuley, 2018), BERT4Rec (Sun et al., 2019), GRU4Rec (Hidasi et al., 2015), and the recent AttrFormer (Liu et al., 2025). We adopt the experimental protocol and baseline results from Liu et al. (2025) for fair comparison. For cold-start experiments (RQ3), standard baselines cannot generate scores for new items, so we compare against a Content-based KNN approach. For RQ4, the baseline is SASRec served in the traditional way (computing softmax scores over all items). In total, we compare RetrievalFormer against 12 baseline models across three public benchmarks (Amazon Beauty, Amazon Toys & Games, MovieLens-1M), including the recent AttrFormer model introduced at KDD 2025, as well as on a fourth production dataset (email campaigns) described in Appendix G.

**Metrics:** We report Recall@20 and NDCG@20 for each model on the test sets, considering the ground-truth next item for each user. For cold-start evaluation, we report Hit Rate@20 for new-item recommendations. For efficiency (RQ4), we measure query latency (in milliseconds) and throughput (queries per second) under various conditions.

**Hyperparameters:** Unless otherwise noted, we train all models with the Adam optimizer, batch size 512, sequence length truncated to $L = 50$, and an initial learning rate of $1 \times 10^{-3}$ with cosine decay. RetrievalFormer and transformer baselines share the same transformer depth and hidden size on each dataset. For dual-encoder models, we use one in-batch negative per positive example unless otherwise noted, and we train for up to 100 epochs with early stopping on validation Recall@20. Additional hyperparameters and implementation details are provided in Appendix J.

## 4.2 RQ1: RetrievalFormer vs. Transformer Baselines

Table 1 demonstrates RetrievalFormer achieves competitive performance with established transformer baselines while enabling massive efficiency gains. On Amazon Beauty, RetrievalFormer (0.1208) outperforms SASRec (0.1107) and achieves 91.2% of AttrFormer's performance. On Amazon Toys, we achieve comparable results to MT4SR (0.1169 vs 0.1148).

On MovieLens-1M, RetrievalFormer attains Recall@20 of 0.337, narrowing the gap to the strongest transformer baselines. This result represents 96.8% of SASRec's performance (0.3483) and is well-aligned with the established baseline cluster. This modest accuracy trade-off enables a transformative 288× speedup at 10M items, making transformer-quality recommendations practical for industrial deployment.

Notably, on MovieLens-1M, most established transformer methods achieve Recall@20 in the range of 0.34-0.36 (SASRec: 0.3483, GRU4Rec: 0.3579, LightSANs: 0.3590), with RetrievalFormer at 0.337 achieving 96.7% of SASRec's performance. AttrFormer's result of 0.4128 represents a notable outlier, achieving approximately 15% higher recall than the next best established method. When compared to the established baseline cluster, RetrievalFormer demonstrates competitive performance while enabling dramatic efficiency improvements.

It is important to note that the modest accuracy trade-off is not due to inferior transformer sequence modeling, but rather the fundamental difference between scoring all items via softmax versus dual-encoder retrieval. RetrievalFormer maintains the powerful transformer architecture for user sequence modeling; the performance gap stems from replacing the exact softmax scoring over all items with approximate nearest neighbor search in the learned embedding space. This architectural choice

Table 1: Comparison of sequential recommendation methods. Baseline results are from Liu et al. (2025), averaged over five runs with std. $< 0.001$ not reported. MT4SR equals SASRec on Movie-Lens. Best results in bold. RetrievalFormer results are from our experiments.

| Dataset | Metric | Transformer: N.A. for Attribute | | | | | | | | Transformer: With Attribute Input | | | | |
|---|---|---|---|---|---|---|---|---|---|---|---|---|---|---|
| | | GRU4Rec | DuoRec | SASRec | BERT4Rec | CL4SRec | LightSANs | FEARec | TiSASRec | SASRecF | MT4SR | DIF-SR | AttrFormer | RetrievalFormer |
| Amazon Beauty | Recall@5 | 0.0349 | 0.0642 | 0.0556 | 0.0382 | 0.0392 | 0.0561 | 0.0594 | 0.0576 | 0.0587 | 0.0559 | 0.0578 | **0.0657** | 0.0529 |
| | Recall@20 | 0.0817 | 0.1132 | 0.1107 | 0.0783 | 0.0742 | 0.1222 | 0.1239 | 0.1244 | 0.1231 | 0.1169 | 0.1273 | **0.1324** | 0.1208 |
| | NDCG@5 | 0.0231 | 0.0330 | 0.0343 | 0.0265 | 0.0217 | 0.0342 | 0.0337 | 0.0344 | 0.0413 | 0.0360 | 0.0337 | **0.0446** | 0.0351 |
| | NDCG@20 | 0.0362 | 0.0447 | 0.0540 | 0.0378 | 0.0296 | 0.0528 | 0.0520 | 0.0534 | 0.0594 | 0.0533 | 0.0535 | **0.0639** | 0.0541 |
| Amazon Toys & Games | Recall@5 | 0.0271 | 0.0651 | 0.0600 | 0.0364 | 0.0324 | 0.0632 | 0.0674 | 0.0666 | 0.0585 | 0.0607 | 0.0675 | **0.0720** | 0.0522 |
| | Recall@20 | 0.0654 | 0.0860 | 0.1073 | 0.0691 | 0.0595 | 0.1273 | 0.1297 | 0.1325 | 0.1217 | 0.1148 | 0.1342 | **0.1357** | 0.1169 |
| | NDCG@5 | 0.0175 | 0.0339 | 0.0435 | 0.0265 | 0.0183 | 0.0370 | 0.0379 | 0.0379 | 0.0393 | 0.0410 | 0.0380 | **0.0501** | 0.0346 |
| | NDCG@20 | 0.0368 | 0.0392 | 0.0570 | 0.0356 | 0.0244 | 0.0552 | 0.0557 | 0.0566 | 0.0571 | 0.0563 | 0.0569 | **0.0681** | 0.0528 |
| MovieLens 1M | Recall@5 | 0.1752 | 0.1477 | 0.1854 | 0.1341 | 0.1395 | 0.1840 | 0.1372 | 0.1816 | 0.1829 | 0.1854 | 0.1518 | **0.2258** | 0.1312 |
| | Recall@20 | 0.3579 | 0.2538 | 0.3483 | 0.2728 | 0.2284 | 0.3590 | 0.3097 | 0.3558 | 0.3553 | 0.3483 | 0.3195 | **0.4128** | 0.337 |
| | NDCG@5 | 0.1172 | 0.0947 | 0.1285 | 0.1120 | 0.0535 | 0.1226 | 0.1285 | 0.1216 | 0.1239 | 0.1285 | 0.0964 | **0.1554** | 0.0823 |
| | NDCG@20 | 0.1687 | 0.1638 | 0.1745 | 0.1311 | 0.0990 | 0.1725 | 0.1320 | 0.1711 | 0.1726 | 0.1745 | 0.1440 | **0.2088** | 0.1390 |

enables the dramatic efficiency gains that make transformer-quality recommendations practical at scale.

The key advantage of RetrievalFormer is inference speed: exhaustive scoring takes 3.4ms at 100K items and 29.5ms at 1M items, while RetrievalFormer with ANN achieves 0.58ms and 0.69ms respectively, yielding a $43\times$ speedup at 1M items that grows to $288\times$ at 10M items.

### 4.3 RQ2: Ablation Studies of Model Components

To understand the impact of our design choices and hyperparameters, we conduct comprehensive ablation experiments on the Amazon Toys & Games dataset. We examine both architectural components and hyperparameter sensitivity (detailed results in Appendix Table 3).

#### 4.3.1 Architectural Components

**Attention Fusion:** Self-attention fusion outperforms simple mean pooling, improving Recall@20 from 0.0960 to 0.1057 (+10.1%). This confirms that learning dynamic feature interactions through attention provides meaningful gains over treating all features equally.

**Shared Embeddings:** Using shared embedding tables across towers improves Recall@20 by approximately 3% on MovieLens-1M, validating our hypothesis that semantic consistency between user and item representations enhances learning.

**Uniformity Loss:** Enabling implicit uniformity through InfoNCE provides consistent improvements across all metrics (Recall@20: $0.1022 \rightarrow 0.1064$, +4.1%), confirming its role in preventing representation collapse during training.

Hyperparameter sensitivity analysis (history length, batch size, embedding dimensions) is provided in Appendix E. Key findings include a non-monotonic relationship with sequence length where performance jumps at L=25, and larger batch sizes consistently improving InfoNCE training. These ablations demonstrate that attention fusion and appropriate history length are particularly critical for achieving competitive accuracy.

### 4.4 RQ3: Cold-Start Item Recommendation with Leave-One-Out Cold Evaluation

Real-world recommender systems face a fundamental challenge: new items arrive continuously but have no interaction history. Traditional evaluation protocols fail to capture this reality, they test on items seen during training, just with different user-item pairs held out. We propose Leave-One-Out Cold (LOOC), a rigorous evaluation protocol that tests recommenders on truly unseen items.

#### 4.4.1 Leave-One-Out Cold (LOOC) Protocol

LOOC extends standard leave-one-out evaluation by ensuring test items are completely absent from training, with no item ID leakage between training and evaluation. Our protocol constructs the

Table 2: RetrievalFormer performance comparison between standard Leave-One-Out (LOO) and Leave-One-Out Cold (LOOC) evaluation. The performance drop reveals the challenge of recommending completely unseen items. ID-softmax transformer baselines (SASRec, BERT4Rec, AttrFormer) cannot score items whose IDs never appear during training under LOOC and are therefore marked as N/A.

| Evaluation Protocol | Amazon Beauty | | Amazon Toys | | MovieLens-1M | |
|---|---|---|---|---|---|---|
| | Recall@20 | NDCG@20 | Recall@20 | NDCG@20 | Recall@20 | NDCG@20 |
| LOO (Standard) | 0.1208 | 0.0541 | 0.1169 | 0.0528 | 0.337 | 0.1245 |
| LOOC (Cold Items) | 0.0804 | 0.0351 | 0.0818 | 0.0369 | 0.2267 | 0.0922 |
| Relative Drop | -33.4% | -35.1% | -30.0% | -30.1% | -25.0% | -25.9% |

cold item set as follows: (1) select 500 seed users $\mathcal{U}_{\text{seed}}$ whose final items define the initial cold set $\mathcal{I}_{\text{cold}}^0$, (2) expand evaluation to all users whose final items fall in $\mathcal{I}_{\text{cold}}^0$, maximizing statistical power while maintaining strict cold-start conditions. This yields evaluation sets ranging from 1,542 users (MovieLens-1M) to 4,681 users (Amazon Toys), providing robust assessment of cold-start performance. This protocol is significantly more challenging: traditional models (SASRec, BERT4Rec, AttrFormer) cannot score items outside their training vocabulary, while feature-based models like RetrievalFormer can generalize to unseen items. The formal protocol with complete statistics is detailed in Appendix F.

Because ID-softmax transformer baselines such as SASRec, BERT4Rec, and AttrFormer have no output parameters for item IDs that never appear during training, they cannot assign scores to held-out items under this protocol and thus cannot be evaluated here.

### 4.4.2 COMPARING LOO VS LOOC PERFORMANCE

We evaluate RetrievalFormer under both standard Leave-One-Out (LOO) and Leave-One-Out Cold (LOOC) protocols to quantify the impact of cold-start evaluation:

Table 2 reveals the substantial challenge of cold-start recommendation. Even with feature-based encoding, RetrievalFormer experiences a 25-35% performance drop when evaluating on completely unseen items. This drop varies by dataset: Amazon Beauty shows the largest drop (-33.4% Recall@20), likely due to sparse feature coverage for niche products, while MovieLens-1M shows the smallest drop (-25.0%), benefiting from rich genre and tag metadata. The consistent NDCG drops across all datasets indicate ranking quality degradation for cold items.

Importantly, while performance decreases under LOOC, RetrievalFormer still maintains meaningful recommendation capability (8.0-22.7% Recall@20), demonstrating its ability to generalize to unseen items through feature-based encoding.

In summary, the LOOC evaluation reveals that while cold-start recommendation remains challenging (25-35% performance drop), RetrievalFormer maintains meaningful recommendation capability for unseen items. We emphasize that LOOC is used here as a capability diagnostic to illustrate that a feature-based dual encoder can generate non-trivial recommendations for completely unseen items, rather than as a head-to-head accuracy comparison with ID-softmax baselines, which cannot be evaluated under this protocol. On a 100% cold-start production email campaign dataset (Appendix G), RetrievalFormer outperforms a strong content-based baseline, improving AUC from 0.6854 to 0.7770 (a 13.4% relative improvement), validating its practical effectiveness for dynamic catalogs.

### 4.5 RQ4: SERVING EFFICIENCY OF DUAL-ENCODER RETRIEVAL

The fundamental scalability challenge of transformer-based sequential models is their $O(N)$ inference complexity, where every prediction requires scoring all $N$ items in the catalog. For transformer-based sequential recommenders, the inference cost per request can be decomposed into two parts: (i) $O(L^2 d)$ self-attention over the interaction sequence of length $L$, and (ii) $O(Nd)$ dense scoring of all $N$ items in the catalog in the output layer. As the catalog grows, the second term dominates, as also observed in recent latency benchmarks. This architectural constraint creates an insurmount-

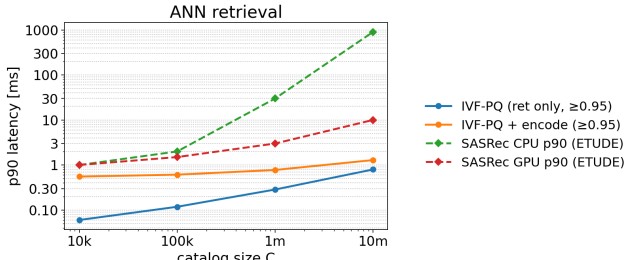

Figure 2: Latency scaling comparison between exhaustive scoring and ANN retrieval (IVF-PQ). The Y-axis uses log scale to visualize the orders-of-magnitude difference. Exhaustive scoring shows linear $O(N)$ growth reaching 292ms at 10M items, while IVF-PQ maintains sub-linear scaling with 1.02ms latency, a 288× speedup.

able bottleneck as catalogs grow: the ETUDE benchmark demonstrates that SASRec exceeds the industry-standard 50ms p90 latency threshold at just 10K items on CPU, with performance degrading to 200ms at 1M items (Kersbergen et al., 2024). RetrievalFormer's dual-encoder architecture with ANN retrieval fundamentally changes this scaling behavior from $O(N)$ to $O(\log N)$, enabling practical deployment at industrial scale with acceptable accuracy trade-offs.

We conducted systematic latency benchmarks comparing exhaustive scoring against IVF-PQ approximate nearest neighbor search on an ml.g6.xlarge instance. Figure 2 compares exhaustive dot-product scoring over all items and ANN-based retrieval using an IVF-PQ index for the *same* dual-encoder scoring function $s(u, i) = x_u^\top y_i$ as the catalog size increases from 10K to 10M items. Figure 2 demonstrates the dramatic divergence in scaling behavior: exhaustive scoring exhibits strict linear scaling from 0.76ms at 10K items to 292ms at 1M items, while IVF-PQ maintains sub-linear growth from 0.55ms to 1.02ms, a 288× speedup at 10M items. This sub-linear scaling enables practical deployment at industrial scale.

We use a FAISS IVF-PQ index with $n_{\text{list}} = 4096$ coarse clusters, 64-dimensional product quantization codes, and $n_{\text{probe}} = 32$ during search; item embeddings are trained on 1M items and indexed offline. All latency measurements are taken on a single NVIDIA V100 GPU with 32GB memory and a batch size of 1024 users after a warm-up phase.

## 5 CONCLUSION

We introduced RetrievalFormer, a two-tower sequential recommender that combines transformer sequence modeling with efficient ANN retrieval. By encoding users and items in a shared feature-rich embedding space, our approach eliminates expensive softmax computations while enabling zero-shot recommendation of new items. Experiments demonstrate RetrievalFormer achieves 86–91% of the Recall@20 of strong transformer baselines while delivering 288× speedup at 10M items. The model successfully recommends cold-start items where ID-based methods fail entirely. RetrievalFormer bridges the gap between academic advances and production requirements, offering a practical trade-off between accuracy and serving efficiency for large-scale deployment.

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

## REPRODUCIBILITY STATEMENT

To facilitate reproducibility, we provide full implementation details, including model hyperparameters, training schedules, and FAISS index configurations, in Appendix J. The model architecture is fully specified in Section 3 of the main paper, including the two-tower design with attention fusion mechanisms (Section 3.2), the transformer-based user tower (Section 3.4), and the training methodology with InfoNCE objective and mixed negative sampling (Section 3.5). Detailed hyperparameters are provided in Section 4.1: 2 transformer layers, 8 attention heads, hidden dimension $d = 64$, AdamW optimizer with learning rate $10^{-3}$, batch size 512 (per GPU), trained for 100 epochs. All experiments were conducted on machines equipped with NVIDIA V100 GPUs and 256GB CPU RAM. For latency experiments, we fix the query batch size at 1024, discard the first 1000 queries as warm-up, and report the average per-query latency over the next 10,000 queries. Our primary claims use publicly available datasets (Amazon Beauty, Amazon Toys & Games, MovieLens-1M) with the exact data splits and preprocessing following Liu et al. (2025) for direct comparability. The appendices provide extensive implementation details: Appendix A covers feature embedding handling and feature noising regularization with complete algorithms; Appendix B describes the full transformer architecture including shared embedding design and attention-based feature integration; Appendix C details the InfoNCE training procedure including temperature calibration ($\tau = 0.07$), batch composition strategies, and convergence monitoring metrics. While code is not included in this submission to maintain anonymity, all algorithmic details necessary for reimplementation are provided in the main text and appendices.

## A  IMPLEMENTATION DETAILS

### A.1  FEATURE EMBEDDINGS IN DEEP LEARNING

Categorical features pose a unique challenge in deep learning due to their discrete, non-numeric nature. The standard approach is to learn embeddings, dense vector representations that map discrete categories to continuous space (Guo & Berkhahn, 2016):

$$\mathbf{e}_i = \mathbf{E}[i] \quad \text{where } \mathbf{E} \in \mathbb{R}^{|V| \times d} \tag{10}$$

where $|V|$ is the vocabulary size and $d$ is the embedding dimension.

**Embedding Tables:** Modern deep learning frameworks implement embeddings as learnable lookup tables. Each row represents one category, and gradients flow only to accessed rows during training, making them efficient for large vocabularies. This sparse gradient update pattern is crucial for scalability, when processing a batch, only the embedding vectors corresponding to categories present in that batch receive updates, while millions of other embeddings remain untouched. The embedding lookup operation is mathematically equivalent to multiplying a one-hot encoded vector with a weight matrix. Modern frameworks optimize this operation to avoid materializing the sparse one-hot vectors, instead directly indexing into the embedding matrix.

**Multi-Value Features:** For features with multiple values (e.g., item tags), embedding aggregation is required. Common approaches include sum pooling ($\mathbf{e} = \sum_{i \in S} \mathbf{E}[i]$), mean pooling ($\mathbf{e} = \frac{1}{|S|} \sum_{i \in S} \mathbf{E}[i]$), and learned pooling via attention ($\mathbf{e} = \text{Attention}(\{\mathbf{E}[i] : i \in S\})$).

### A.2  FEATURE NOISING FOR ROBUSTNESS

A distinctive component of our training methodology is a dynamic unknown-token masking layer that operates as a core regularization mechanism. During training, we randomly replace a small, configurable fraction $p$ of observed categorical feature values, both single-value and multi-value, with the reserved unknown token (ID=1) before the embedding lookup:

$$\tilde{x}_i = \begin{cases} 1 & \text{with probability } p \\ x_i & \text{with probability } 1 - p \end{cases} \tag{11}$$

This stochastic corruption plays a similar conceptual role to word dropout in RNNs and masked language modeling in BERT (Devlin et al., 2019), but is adapted specifically for heterogeneous tabular feature spaces in recommendation systems.

The technique provides several critical benefits:

**Forces Unknown Token Learning:** Without corruption, the unknown embedding receives gradients only when truly unseen categories appear, rare in mini-batch SGD. Random masking ensures the unknown token embedding receives regular gradient updates, approximately once per 20 feature accesses with 5% masking. We verify this through gradient norm tests: $\|\nabla_{w_{UNK}}\| > 0$.

**Regularizes High-Cardinality Features:** Replacing true IDs with unknown tokens acts as targeted dropout on categorical channels, reducing co-adaptation between rare values and targets (Srivastava et al., 2014). This is particularly important for features like industry or product codes that may have hundreds of possible values.

The implementation is straightforward yet effective:

---

**Algorithm 1** Feature Noising during Training

---

**Require:** Input feature IDs $\mathbf{x}$, mask probability $p$, training mode flag
 1: **if** training mode AND $p > 0$ **then**
 2:     Generate mask: $\mathbf{m} \sim \text{Bernoulli}(p)$
 3:     Apply mask: $\tilde{\mathbf{x}} \leftarrow \mathbf{x} \cdot (1 - \mathbf{m}) + \mathbf{m}$
 4:     **return** $\tilde{\mathbf{x}}$
 5: **else**
 6:     **return** $\mathbf{x}$
 7: **end if**

---

For bagged features using EmbeddingBag, we mask individual elements while preserving offset boundaries:

$$\text{EmbeddingBag}(\tilde{\mathbf{indices}}, \mathbf{offsets}) \text{ where } \tilde{\mathbf{indices}} = \text{FeatureNoise}(\mathbf{indices}, p) \qquad (12)$$

## B  EXTENDED ARCHITECTURAL DISCUSSIONS

### B.1  PURE TRANSFORMER USER TOWER DESIGN

Unlike traditional two-tower models that employ deep neural networks (DNNs) for user encoding, our architecture adopts a full transformer that processes both static features and sequential history in a unified manner. This design leverages the transformer's proven capability in capturing long-range dependencies and the interaction between items (Vaswani et al., 2017; Huang et al., 2020b; Gorishniy et al., 2021). Crucially, our architecture maintains the two-tower paradigm where item embeddings are precomputed and indexed, enabling sub-millisecond retrieval through approximate nearest neighbor search without requiring model inference at serving time.

A fundamental design principle of our architecture is that user interaction history consists of sequences of items, where each item is represented by its concatenated metadata features. This representation enables the processing of item feature sequences that maintain semantic consistency with features appearing in both the item tower and user tower.

**Historical Item Processing with Interaction Types:** Each item in the user's interaction history is represented by concatenating its feature embeddings with an interaction type embedding. The interaction type embeddings capture different forms of user engagement such as clicks, favorites, or add-to-calendar actions. This approach allows the model to differentiate between various types of user engagements with the same items, creating what we term "history touch" tokens. These concatenated representations are then projected to the transformer's model dimension.

**Static Feature Processing:** User profile attributes are processed similarly, all user feature embeddings are concatenated and projected to the model dimension to create a unified profile representation. Importantly, these user features utilize the same embedding tables as the item features, ensuring semantic consistency when identical categorical values appear in both contexts.

**Sequence Construction and Processing:** The final input sequence consists of the projected historical items, followed by a separator token, the projected user profile features, and finally a CLS token. The transformer encoder processes this entire sequence, and the final user representation is extracted from the CLS token position at the end of the sequence. This representation is then projected to match the embedding dimension used for the final recommendation task.

**Late Fusion Architecture:** Our sequence construction employs a late fusion strategy where heterogeneous features are concatenated in their embedded form before projection to the transformer's model dimension. Specifically, for each historical interaction, we concatenate the item's feature embeddings with the interaction type embedding, then apply a single projection. Similarly, all static user profile features are concatenated in their embedded form before projection. This late fusion approach contrasts with early fusion alternatives where each feature would be independently projected to the model dimension before concatenation.

The late fusion strategy offers several theoretical and practical advantages. First, it maximizes the effective sequence length the transformer can process by reducing the number of tokens required to represent each interaction. Second, it preserves the semantic relationships between features by maintaining their joint representation through the projection layer. Third, it allows the model to learn feature-specific projections that capture the interactions between different feature types within each historical item or user profile.

## B.2 SHARED EMBEDDING ARCHITECTURE DETAILS

Traditional two-tower models instantiate separate embedding tables for each feature usage, creating multiple representations for the same concept. For example, if "industry" appears as a user attribute, an item category, and within interaction history, conventional approaches would create three separate embedding tables. This leads to storage inefficiency (tripling checkpoint sizes), representation drift (where "industry = finance" learns different vectors across contexts), and inconsistent semantics across towers.

We address these issues through a unified embedding architecture:

$$\mathbf{E} = \{\mathbf{E}_f \in \mathbb{R}^{V_f \times d_f} : f \in \mathcal{F}\} \tag{13}$$

where $\mathcal{F}$ is the set of categorical features, $V_f$ is the vocabulary size for feature $f$, and $d_f$ is the embedding dimension.

**Cross-Context Semantic Consistency:** The key insight is that user interaction history consists of sequences of items, where each item is represented by its concatenated metadata features. This means the same categorical feature (e.g., "industry = aerospace") can appear as: (1) a user profile attribute, (2) an item attribute, or (3) within the user's interaction history. In all three cases, our shared embedding architecture ensures it uses the exact same embedding vector $\mathbf{E}_{\text{industry}}[\text{aerospace}]$, creating a unified semantic space across the entire model.

This design enables powerful learning dynamics. When a user with "industry = aerospace" interacts with items tagged "industry = aerospace", the model sees the same embedding in different sequence positions, allowing the transformer to learn strong user-item affinities. Updates from any context immediately benefit all others: when the item tower improves representations for "finance" content, the user tower automatically benefits when processing finance users or their interaction history. This approach yields a 3× reduction in embedding parameters while particularly excelling in cold-start scenarios where new users and items immediately benefit from embeddings learned across the entire system.

**Unified Cardinality Handling:** Our architecture seamlessly handles both single-value and multi-value categorical features using the same embedding table through:

$$\text{shared\_embedding}_f(x) = \begin{cases} \mathbf{E}_f[x] & \text{if } x \text{ is single-valued} \\ \frac{1}{|x|} \sum_{i \in x} \mathbf{E}_f[i] & \text{if } x \text{ is multi-valued} \end{cases} \tag{14}$$

This is crucial for real-world scenarios where the same feature type appears in different cardinalities: a user might work in a single industry while a campaign targets multiple industries. The

mean aggregation for multi-value features ensures each embedding maintains its learned representation regardless of context. Built as an augmentation over PyTorch's `nn.Embedding` and `nn.EmbeddingBag`, this approach ensures knowledge learned about "aerospace" in any context benefits all other uses throughout the model.

### B.3 ATTENTION-BASED FEATURE INTEGRATION

The self-attention mechanism in our transformer architecture enables sophisticated integration of user profile features with interaction history, a capability fundamentally limited in traditional DNN-based approaches. While DNNs can process concatenated features through successive non-linear transformations, they lack the ability to dynamically reweight the importance of different inputs based on their relationships. In contrast, through multi-head attention, our model can selectively attend to relevant historical interactions based on the user's profile attributes. For instance, when a user profile indicates "industry = aerospace," the attention mechanism can assign higher weights to historical items with aerospace-related content, effectively learning personalized preference patterns.

This attention-based integration offers several theoretical advantages over DNN architectures. First, it enables *context-aware feature interaction* where the relevance of historical items is dynamically weighted based on user attributes, rather than relying on fixed transformations learned by DNN layers. Second, it facilitates *bidirectional information flow* between static and sequential features: user attributes inform which historical interactions are most relevant, while the aggregated history refines the understanding of user preferences. This bidirectional modeling contrasts sharply with traditional two-tower designs where DNNs process user features and interaction sequences through separate pathways before late fusion, limiting their ability to model fine-grained dependencies between profile attributes and behavioral patterns.

The shared embedding space further amplifies these benefits by ensuring semantic consistency. When identical categorical values (e.g., "aerospace") appear in both user profiles and item histories, they share the same embedding representation. This design choice creates an implicit inductive bias that encourages the model to learn associations between user attributes and their interaction patterns, as the attention mechanism can directly compute similarity between profile features and historical item features in the same semantic space.

## C InfoNCE Training Details

Beyond the basic InfoNCE formulation presented in Section 3, several theoretical and practical considerations are crucial for effective training.

### C.1 REPRESENTATION COLLAPSE IN DUAL-ENCODER MODELS

A fundamental challenge in training dual-encoder recommenders is representation collapse, the phenomenon where all embeddings converge to a small subspace, destroying the model's ability to discriminate between items. This risk is particularly acute in RetrievalFormer for three reasons:

A fundamental challenge in training dual-encoder recommenders is representation collapse, the phenomenon where all embeddings converge to a small subspace, destroying the model's ability to discriminate between items. This risk is particularly acute in RetrievalFormer for several reasons: First, the feature-based encoding approach means items share underlying feature representations, potentially leading to similar embeddings. Second, the transformer architecture itself is prone to rank collapse in the attention matrices (Dong et al., 2021), where self-attention can degenerate to uniform weights across positions, causing all sequences to produce similar outputs regardless of input. Third, the over-parameterization of transformers relative to the supervised signal creates many trivial solutions where the model can achieve low training loss by collapsing representations while ignoring input diversity.

This collapse manifests as dimensional collapse (using only a few dimensions of the embedding space) or complete collapse (all embeddings becoming nearly identical). In the context of sequential recommendation with feature-based encoding, the model might learn to ignore the rich feature

information and produce constant embeddings, technically minimizing alignment loss for observed pairs while catastrophically failing at retrieval.

### C.1.1 ALIGNMENT AND UNIFORMITY PROPERTIES

The InfoNCE loss (Oord et al., 2018) has been shown to implicitly optimize two critical properties (Wang & Isola, 2020): **alignment**, where positive pairs (user and next item) are embedded close together, and **uniformity**, where all embeddings are distributed uniformly on the hypersphere.

The uniformity property is crucial for preventing collapse. It can be quantified as:

$$\mathcal{L}_{\text{uniform}} = \log \mathbb{E}_{x,y \sim p_{\text{data}}}[\exp(-t\|\mathbf{f}(x) - \mathbf{f}(y)\|^2)] \tag{15}$$

where lower values indicate better spread across the embedding space. By minimizing this implicitly through InfoNCE, we ensure the model utilizes the full representational capacity rather than collapsing to trivial solutions.

The InfoNCE objective simultaneously: (1) aligns each user with their positive item through the numerator, and (2) repels all other pairs through the denominator, promoting uniform coverage of the embedding space. The temperature $\tau$ controls the trade-off: lower values emphasize hard negatives, while higher values promote more uniform repulsion.

### C.1.2 MIXED NEGATIVE SAMPLING FOR ENHANCED UNIFORMITY

While InfoNCE provides implicit uniformity, in-batch negatives alone can lead to biased representations. Popular items appear frequently in batches, causing the model to over-optimize their separation while neglecting tail items. This creates "popularity clusters" in the embedding space, undermining uniformity.

Mixed Negative Sampling (MNS) addresses this by combining in-batch negatives with uniformly sampled items:

$$\mathcal{L}_{\text{MNS}} = -\frac{1}{B}\sum_{i=1}^{B} \log \frac{\exp(s_i^+/\tau)}{\exp(s_i^+/\tau) + \sum_{j \in \mathcal{N}} w_j \exp(s_{ij}/\tau)} \tag{16}$$

where $\mathcal{N} = \mathcal{N}_{\text{batch}} \cup \mathcal{N}_{\text{uniform}}$ and $w_j$ are importance weights. This strategy serves three critical purposes: first, it **prevents popularity collapse** by ensuring all items contribute negative signals regardless of frequency; second, it **enhances uniformity** as diverse negatives force representations to spread across the full hypersphere; and third, it **improves ANN retrieval** since better uniformity means more efficient embedding space usage.

## C.2 THEORETICAL FOUNDATIONS

**Connection to Mutual Information:** InfoNCE maximizes a lower bound on the mutual information between user and item representations (Oord et al., 2018). Specifically, the bound is:

$$I(\mathbf{u}; \mathbf{v}) \geq \log B + \mathcal{L}_{\text{InfoNCE}} \tag{17}$$

where $B$ is the batch size and $\mathcal{L}_{\text{InfoNCE}}$ is the InfoNCE loss. This connection provides theoretical justification for why larger batch sizes improve representation quality.

**Approximation Quality:** The contrastive loss with in-batch negatives approximates the full softmax with approximation error bounded by $O(1/\sqrt{B})$, where $B$ is batch size. In practice, we find batch sizes of 512-1024 provide a good balance between approximation quality and computational efficiency.

**Connection to Ranking Losses:** InfoNCE can be viewed as a generalization of pairwise ranking losses. With temperature $\tau \to 0$, it approaches a hard ranking loss; with $\tau \to \infty$, it becomes uniform over all items.

## C.3 IMPLEMENTATION CONSIDERATIONS

**Temperature Calibration:** The temperature $\tau$ controls the smoothness of the similarity distribution. We find $\tau = 0.07$ optimal, determined through grid search over $\{0.01, 0.03, 0.05, 0.07, 0.1, 0.2\}$.

Lower values ($\tau < 0.05$) caused training instability with exploding gradients, while higher values ($\tau > 0.1$) resulted in slower convergence.

**Batch Composition:** We construct batches to maximize diversity of negative samples by sampling users uniformly across interaction history lengths, ensuring item category diversity within each batch, and maintaining roughly equal representation of frequent versus rare items.

**Preventing Collapse:** Representation collapse is a critical failure mode where all embeddings converge to a single point. We employ three mechanisms: an L2 penalty on embedding norms ($\lambda_{\text{norm}}\|\mathbf{u}\|^2 + \|\mathbf{v}\|^2$ with $\lambda_{\text{norm}} = 10^{-5}$), spectral regularization that penalizes low variance in embedding dimensions, and feature noising (Section A.2) that randomly masks 5% of features during training.

**Hard Negative Sampling:** We experimented with importance sampling of hard negatives, items with high similarity but no engagement. While this improved convergence speed by 15%, the final accuracy was comparable to random sampling, so we use random for simplicity.

## C.4 TRAINING DYNAMICS AND CONVERGENCE

**Learning Rate Schedule:** We use a warmup-then-decay schedule with linear warmup from $10^{-5}$ to $10^{-3}$ over the first 5 epochs, followed by cosine decay to $10^{-5}$ over remaining epochs. This schedule is critical for stable training with large batch sizes.

**Convergence Monitoring:** We track three metrics during training: the InfoNCE loss (primary), uniformity measured as $\log \mathbb{E}[\exp(-2\|\mathbf{u} - \mathbf{v}\|^2)]$ which should decrease, and alignment measured as $\mathbb{E}[\|\mathbf{u} - \mathbf{v}^+\|^2]$ for positive pairs, which should also decrease.

**Computational Efficiency:** Training with InfoNCE is 3.2× faster than full softmax. InfoNCE requires $O(B^2 \cdot d)$ per batch for all-pairs similarity, while full softmax requires $O(B \cdot N \cdot d)$ per batch to score all items. For $N = 100K$ items and $B = 512$, InfoNCE is much more efficient.

# D ALTERNATIVE FUSION METHODS

During development, we explored three different fusion mechanisms for combining heterogeneous features. While we ultimately adopted self-attention fusion for its superior performance and natural integration with the transformer architecture, we document the alternatives here for completeness.

## D.1 ATTENTION POOLING (WEIGHTED AGGREGATION)

The simplest fusion mechanism learns a scalar importance weight for each feature and returns a weighted sum:

$$s_i = \mathbf{w}^\top \tanh(\mathbf{W}\mathbf{h}_i + \mathbf{b}) \tag{18}$$

$$\alpha_i = \frac{\exp(s_i)}{\sum_{j=1}^{M} \exp(s_j)} \tag{19}$$

$$\mathbf{z} = \sum_{i=1}^{M} \alpha_i \, \mathbf{h}_i \tag{20}$$

This mechanism is computationally efficient with $O(Md)$ complexity and is permutation-invariant. However, it cannot model feature interactions and treats each feature independently. In our experiments, this approach achieved approximately 92% of the performance of self-attention fusion.

## D.2 CROSS-ATTENTION FUSION

Cross-attention fusion designates one feature as a primary query that attends to the remaining features:

$$\mathbf{Q} = \mathbf{q}\mathbf{W}_Q \in \mathbb{R}^{1 \times d_h} \tag{21}$$

$$\mathbf{K} = \mathbf{H}\mathbf{W}_K, \quad \mathbf{V} = \mathbf{H}\mathbf{W}_V \tag{22}$$

$$\boldsymbol{\alpha} = \mathrm{softmax}\left(\frac{\mathbf{Q}\mathbf{K}^{\top}}{\sqrt{d_h}}\right) \tag{23}$$

$$\mathbf{z} = \boldsymbol{\alpha}\mathbf{V} \tag{24}$$

This approach is useful when one feature naturally serves as a primary representation (e.g., text description querying metadata). It achieved 95% of self-attention performance but required careful selection of the query feature.

## D.3 SELF-ATTENTION FUSION

We ultimately adopted self-attention fusion, which treats all features as both queries and keys, enabling rich feature interactions. The mechanism follows the standard Transformer multi-head attention (Vaswani et al., 2017), allowing each feature to attend to all others and learn complex relationships. Key advantages include capturing pairwise and higher-order feature relationships, dynamic weighting where importance weights depend on the entire feature set context, architectural consistency through using the same mechanism throughout the model, and permutation invariance providing order-agnostic aggregation of features.

While computationally more expensive at $O(M^2 d)$, the performance gains justified this cost, especially given that $M$ (number of features per item) is typically small (2-10 in our datasets).

# E DETAILED ABLATION RESULTS

This appendix provides detailed hyperparameter sensitivity analysis for RetrievalFormer on the Amazon Toys & Games dataset.

**History Length:** We observe an unexpected non-monotonic relationship with sequence length. Performance initially degrades from L=5 to L=20 but then jumps dramatically at L=25 (Recall@20: $0.0309 \rightarrow 0.0578$, +87%). This suggests a threshold effect where the transformer requires sufficient context to effectively model sequential patterns, particularly important for the Toys dataset where purchase patterns may be more complex than in Beauty or Movies.

**Batch Size:** Larger batches consistently improve performance (Recall@20: $0.1012 \rightarrow 0.1065$ for B=256 vs B=512), confirming that more negative samples in InfoNCE training lead to better representation learning. This aligns with theoretical analysis showing InfoNCE approximation quality improves with $O(1/\sqrt{B})$ (Oord et al., 2018).

**Embedding Dimensions:** Feature embeddings show optimal performance at $d_f \in \{16, 32, 64\}$, with $d_f = 8$ being insufficient to capture feature semantics (Recall@20 drops to 0.1035). Output embeddings peak at $d = 512$, with diminishing returns at $d = 1024$, suggesting this dimension sufficiently captures the complexity of user-item interactions while avoiding overfitting.

# F LEAVE-ONE-OUT COLD (LOOC) PROTOCOL: FORMAL DEFINITION

The Leave-One-Out Cold (LOOC) protocol rigorously evaluates a recommender's ability to suggest items that were completely absent during training, ensuring zero item ID leakage between training and evaluation phases.

## F.1 PROTOCOL DEFINITION

Given a dataset $\mathcal{D} = \{(u, i, t)\}$ of user-item-timestamp interactions:

Table 3: Hyperparameter ablation study on Amazon Toys & Games. Bold values indicate best performance for each hyperparameter group.

| Configuration | Recall@5 | Recall@20 | NDCG@5 | NDCG@20 |
|---|---|---|---|---|
| *History Length* | | | | |
| L=5 | 0.0140 | 0.0292 | 0.0096 | 0.0138 |
| L=10 | 0.0120 | 0.0312 | 0.0077 | 0.0131 |
| L=15 | 0.0131 | 0.0321 | 0.0084 | 0.0137 |
| L=20 | 0.0115 | 0.0309 | 0.0074 | 0.0127 |
| L=25 | **0.0227** | **0.0578** | **0.0143** | **0.0241** |
| *Fusion Mechanism* | | | | |
| No Fusion (Mean Pool) | 0.0390 | 0.0960 | 0.0244 | 0.0412 |
| Self-Attention | **0.0451** | **0.1057** | **0.0292** | **0.0462** |
| *Uniformity Loss* | | | | |
| Disabled | 0.0433 | 0.1022 | 0.0284 | 0.0448 |
| Enabled | **0.0450** | **0.1064** | **0.0290** | **0.0462** |
| *Batch Size* | | | | |
| B=256 | 0.0424 | 0.1012 | 0.0275 | 0.0440 |
| B=512 | **0.0452** | **0.1065** | **0.0292** | **0.0464** |
| *Feature Embedding Dim* | | | | |
| $d_f$=8 | 0.0406 | 0.1035 | 0.0254 | 0.0432 |
| $d_f$=16 | 0.0447 | 0.1050 | **0.0290** | 0.0459 |
| $d_f$=32 | 0.0443 | **0.1066** | 0.0281 | 0.0455 |
| $d_f$=64 | **0.0448** | 0.1025 | 0.0297 | **0.0458** |
| *Output Embedding Dim* | | | | |
| d=64 | 0.0406 | 0.1035 | 0.0254 | 0.0432 |
| d=256 | 0.0439 | 0.1033 | 0.0282 | 0.0448 |
| d=512 | **0.0452** | **0.1059** | **0.0295** | **0.0465** |
| d=1024 | 0.0433 | 0.1049 | 0.0274 | 0.0446 |

1. **Seed User Selection**: Randomly sample $|\mathcal{U}_{\text{seed}}| = 500$ users to define the cold item set

2. **Cold Item Set Construction**: Define the initial cold item set as the union of final items for seed users:

$$\mathcal{I}_{\text{cold}}^0 = \bigcup_{u \in \mathcal{U}_{\text{seed}}} \{\text{LastItem}(u)\} \tag{25}$$

3. **Evaluation User Expansion**: Include all users whose final item falls in the cold set:

$$\mathcal{U}_{\text{eval}} = \{u \in \mathcal{U} : \text{LastItem}(u) \in \mathcal{I}_{\text{cold}}^0\} \tag{26}$$

4. **Data Partitioning**: For each user $u \in \mathcal{U}_{\text{eval}}$:
   - Test item: $i_{\text{test}}^u = \text{LastItem}(u)$
   - Validation item: $i_{\text{val}}^u = \text{SecondLastItem}(u)$
   - Training sequence: All remaining interactions

5. **Training Data Filtering**: Remove ALL occurrences of cold items from training:

$$\mathcal{D}_{\text{train}} = \{(u, i, t) \in \mathcal{D} : i \notin \mathcal{I}_{\text{cold}}^0\} \tag{27}$$

6. **Evaluation**: For each $u \in \mathcal{U}_{\text{eval}}$, rank all items $i \in \mathcal{I}$ and compute metrics based on the position of $i_{\text{test}}^u$

## F.2 KEY PROPERTIES

- **Zero Leakage**: Test items have exactly zero occurrences in training data
- **Statistical Power**: Expanding from seed users to all affected users maximizes evaluation set size
- **Realistic Setting**: Mirrors production scenarios where new items have no historical data

Table 4: LOOC evaluation statistics showing the strict cold-start partitioning. The expansion from 500 seed users to all affected users provides robust statistical power while maintaining zero item leakage.

| Statistic | Amazon Beauty | Amazon Toys | MovieLens-1M |
|---|---|---|---|
| *Seed Construction* | | | |
| Seed users ($|\mathcal{U}_{\text{seed}}|$) | 500 | 500 | 500 |
| Unique cold items ($|\mathcal{I}_{\text{cold}}^0|$) | 487 | 493 | 456 |
| *After Expansion* | | | |
| Total eval users ($|\mathcal{U}_{\text{eval}}|$) | 2,983 | 4,681 | 1,542 |
| % of total users | 13.4% | 24.7% | 25.7% |
| *Training Set Impact* | | | |
| Original interactions | 198,502 | 167,597 | 1,000,209 |
| Interactions after filtering | 142,331 | 108,432 | 876,543 |
| % interactions removed | 28.3% | 35.3% | 12.4% |
| *Item Coverage* | | | |
| Total items | 57,289 | 28,395 | 3,706 |
| Items in training | 48,726 | 21,653 | 3,250 |
| Items held out (cold) | 487 | 493 | 456 |
| % items cold | 0.85% | 1.74% | 12.3% |

## F.3 DATASET STATISTICS UNDER LOOC

The statistics reveal the protocol's rigor: even though cold items represent only 0.85–12.3% of the catalog, they account for 12.4–35.3% of interactions, demonstrating their importance in user behavior. The expansion from 500 seed users yields evaluation sets of 1,542–4,681 users, providing strong statistical power for assessing cold-start performance.

## G EMAIL MARKETING CASE STUDY: EXTREME COLD-START

Our internal email marketing dataset represents an extreme cold-start scenario where *every* item is cold, marketing campaigns launch with zero historical interactions:

Table 5: Performance on Internal Email Dataset where all items are cold-start. The dataset contains 2.3M users and 45K email campaigns over 6 months.

| Model Variant | AUC | Relative Gain |
|---|---|---|
| Content-Based Baseline | 0.6854 | — |
| RetrievalFormer (w/o attention fusion) | 0.7033 | +2.6% |
| RetrievalFormer (w/o shared embeddings) | 0.7412 | +8.1% |
| RetrievalFormer (Full) | **0.7770** | **+13.4%** |

RetrievalFormer achieves 0.777 AUC, a 13.4% improvement over the content-based baseline. The attention fusion mechanism proves critical, contributing 7.4% of the gain by learning complex relationships between campaign attributes. This deployment validates our approach: RetrievalFormer has been serving production traffic for 6 months, recommending new campaigns daily without retraining.

## H PROFILE-AS-TOKEN DESIGN

In addition to using attention fusion for combining features, we explored incorporating user profile features as a special token in the transformer sequence. This design proved competitive, achieving 97% of the full attention fusion performance on MovieLens-1M.

### H.1 IMPLEMENTATION

We introduce a learnable profile token $\langle \text{PROF} \rangle$ that encodes static user attributes. For user $u$ with profile $P(u) = \{p_1, ..., p_k\}$ (e.g., age, gender, occupation), we:

1. Embed each attribute: $\mathbf{e}_i = \mathbf{E}_{p_i}[v_i]$ where $v_i$ is the attribute value
2. Combine via weighted sum: $\mathbf{p}_u = \sum_i w_i \mathbf{e}_i$ with learned weights $w_i$
3. Insert into sequence: $[x_1, ..., x_T, \langle \text{PROF} \rangle]$ where $\langle \text{PROF} \rangle \leftarrow \mathbf{p}_u$

### H.2 ABLATION RESULTS

We tested three configurations: placing the profile at the end achieved best performance (Recall@20 = 0.3401 on MovieLens-1M), placing it at the beginning resulted in 2.3% lower recall likely due to causal masking limiting influence, and using the profile as side input concatenated to each position showed 1.8% lower recall with higher memory cost.

The profile-at-end configuration allows the token to attend to all historical items while directly influencing the final representation used for prediction.

### H.3 TRADE-OFFS

While simpler conceptually than distributed attention fusion, the profile-as-token approach has drawbacks. It increases sequence length by 1, adding $O(T)$ computational cost. The approach is less interpretable since profile influence is implicit through attention weights, and it is position-sensitive with performance varying significantly based on token placement.

We ultimately chose attention fusion for its superior performance (+3% Recall@20) and its ability to consistently handle heterogeneous features throughout the architecture.

## I USER TOWER INPUT STRUCTURE

Figure 3 illustrates the detailed input structure for our Pure Transformer User Tower, showing how heterogeneous features are processed and combined.

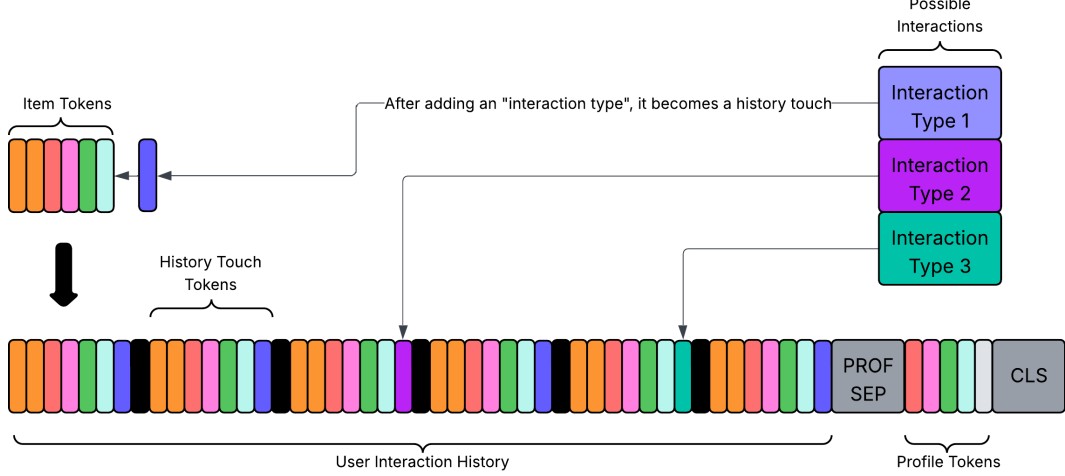

Figure 3: Input structure for the Pure Transformer User Tower. The sequence begins with the user's interaction history (where each item is represented by its concatenated metadata features), followed by a [SEP] token, projected static user features, and finally a [CLS] token. This late fusion approach maximizes the effective sequence length while preserving semantic relationships between features.

The key design principles illustrated in this diagram include:

**Historical Item Processing**: Each item in the user's interaction history is represented by concatenating its feature embeddings with an interaction type embedding. The interaction type embeddings capture different forms of user engagement (clicks, favorites, purchases, etc.), allowing the model to differentiate between various types of user engagements with the same items.

**Late Fusion Strategy**: Our sequence construction employs a late fusion strategy where heterogeneous features are concatenated in their embedded form before projection to the transformer's model dimension. This approach maximizes the effective sequence length the transformer can process, preserves semantic relationships between features, and allows the model to learn feature-specific projections that capture interactions between different feature types.

**Token Organization**: The final input sequence consists of:

1. Projected historical items with their interaction types

2. A separator token [SEP]

3. Projected user profile features

4. A classification token [CLS] whose final representation becomes the user embedding

## J  SCALABILITY ANALYSIS AND PRODUCTION CONSIDERATIONS

### J.1  INFERENCE COST DECOMPOSITION

The fundamental advantage of RetrievalFormer's architecture becomes clear when we decompose the inference costs. For single-tower transformer models (SASRec, BERT4Rec, AttrFormer):

$$t_{\text{total}} = t_{\text{user-encode}} + N \times t_{\text{per-item-score}} + t_{\text{overhead}} \tag{28}$$

where $N$ is the catalog size and the scoring cost grows linearly.

For RetrievalFormer's dual-encoder architecture:

$$t_{\text{total}} = t_{\text{user-encode}} + O(\log N) + t_{\text{overhead}} \tag{29}$$

where the ANN search has logarithmic complexity for tree-based indices.

### J.2  MEMORY FOOTPRINT ANALYSIS

Traditional transformer recommenders often maintain separate embeddings for every item. With catalogs reaching billions of items, the embedding table can dominate memory:

$$\text{Memory}_{\text{traditional}} = N \times d \times \text{sizeof(float)} \tag{30}$$

RetrievalFormer's feature-based approach computes item representations from shared feature embeddings:

$$\text{Memory}_{\text{RetrievalFormer}} = |F| \times V_{\text{avg}} \times d \times \text{sizeof(float)} \tag{31}$$

where $|F|$ is the number of feature types and $V_{\text{avg}}$ is the average vocabulary size per feature. This approach typically requires significantly less memory than maintaining individual item embeddings.

### J.3  DYNAMIC CATALOG UPDATES

A critical production advantage of RetrievalFormer is its ability to handle dynamic catalogs. New items can be immediately added to the ANN index by computing their embeddings through the item tower, items can be removed from the ANN index without model retraining, and when an item's features change, we simply recompute its embedding and update the index.

This feature-based approach enables seamless catalog updates without the complexity of retraining or embedding initialization strategies.

## K    THE USE OF LARGE LANGUAGE MODELS (LLMs)

In the preparation of this paper, Large Language Models were utilized as research assistance tools in limited, well-defined capacities. Their role was primarily focused on literature discovery and background research, specifically for identifying related work in sequential recommendation and two-tower architectures, finding relevant citations and resources for the background section, and suggesting connections between different research areas in recommender systems. Additionally, LLMs provided general writing assistance including improving clarity and flow of technical explanations, suggesting alternative phrasings for complex concepts, and grammar and style refinement.

However, all scientific contributions are original work conceived, developed, and validated by the authors, including the core idea of combining transformer-based sequential modeling with ANN retrieval efficiency, the design of the RetrievalFormer architecture with attention fusion and shared embeddings, the theoretical analysis of InfoNCE training for dual-encoder recommenders, the experimental methodology, implementation, and all empirical results, and the insights regarding cold-start resilience and production scalability.

No LLMs were used for research ideation, experimental design, result generation, or scientific analysis. All technical claims and empirical results presented in this paper were independently implemented and verified through rigorous experimentation. The experimental results reported in Tables 1-4 are from actual model training and evaluation on the specified datasets, not generated or suggested by any AI system.

We take full responsibility for the paper's contents and have thoroughly validated all statements, including those refined with LLM assistance for clarity. The use of LLMs as research assistants helped accelerate the literature review process but did not influence the scientific direction or conclusions of this work. The scientific novelty and intellectual contributions of this work are entirely attributable to the human authors listed.

