# OpenReview forum: "RETRIEVALFORMER: TRANSFORMER-QUALITY RECOMMENDATIONS WITH EFFICIENT ANN RETRIEVAL AND COLD-START RESILIENCE"
_ICLR.cc/2026/Conference — Submitted to ICLR 2026_

### Official Review · Reviewer_r8bb · 2025-10-30

**Soundness:** 1
**Presentation:** 1
**Contribution:** 2
**Rating:** 2
**Confidence:** 4

**Summary:**

The authors propose RetrievalFormer, it is a two-tower model that combines transformer-based sequence modeling with efficient ANN-based retrieval for sequential recommendation. It uses shared embeddings and a multimodal feature encoder to achieve competitive accuracy while enabling fast inference and handling item cold-start. Experiments on Amazon and MovieLens datasets show 86–91% recall of transformer baselines with up to 288× speedup, along with effective recommendation of unseen items.

**Strengths:**

The authors conducted numerous experiments to verify the effectiveness of the method.

**Weaknesses:**

The first argument posited by this paper is that existing transformer-based sequential recommendation models incur high computational costs. However, the explanation of what causes such high costs remains unclear. Is it due to the quadratic complexity of the self-attention mechanism in the transformer architecture? What are the existing solutions proposed to mitigate this issue?

The second argument claims that classical transformers struggle with the item cold-start problem. To the best of my knowledge, this issue is primarily data-related rather than model-related, as it stems from insufficient interaction history for new items.

The authors did not adequately elaborate on how their proposed method addresses the aforementioned issues. In lines 50–60, they devote extensive space to describing implementation details, yet these descriptions fail to substantiate their earlier claims. There is a lack of evidence demonstrating that their approach effectively tackles the efficiency issue of transformers or the cold-start problem.

Moreover, the paper appears hastily prepared and suffers from multiple technical and presentational shortcomings. Below are several examples:

1. The title seems unusual—what does “ANN” refer to? It is not a standard academically recognized term in this context.

2. Question marks appear on lines 260 and 798, and an unclear character “L” is found on line 363.

3. Table 2 exceeds the prescribed layout boundaries.

4. The term “transformer accuracy” is ambiguous and undefined.

In summary, while the paper introduces numerous technical implementation details, it provides no compelling evidence to support its two central claims regarding efficiency and cold-start performance. Additionally, the overall presentation is unpolished and contains numerous errors. In my assessment, the manuscript requires substantial further work before it can be considered acceptable. Therefore, I strongly recommend rejection.

**Questions:**

See weaknesses.

---

> ### Author Response · Authors · 2025-12-01
> **Response to Reviewer r8bb - 1**
>
> We thank the reviewer for the candid and detailed review and for noting that we conduct numerous experiments. Below we respond to each of the main concerns.
>
> Computational cost and existing solutions
>
> Reviewer: “Claims that existing Transformer-based sequential recommendation models incur high computational costs, but provides no clear explanation of what causes these costs … and omits discussion of existing solutions.”
>
> Our clarification: Our efficiency claims target a specific bottleneck in large-catalog sequential recommendation: scoring all N items in the output layer. In standard transformer recommenders (e.g., SASRec, BERT4Rec), inference cost has two main parts: (i) self-attention over the sequence, and (ii) an ID-softmax head that computes logits over all items. When N is large, the O(N·d) output layer typically dominates the O(L²·d) attention cost. RetrievalFormer changes the formulation: we explicitly use a dual-encoder architecture (transformer user tower + item tower) and serve recommendations by dot-product similarity plus ANN search over item embeddings, instead of softmax over all IDs. The reported 288× speedup at 10M items compares exhaustive dot-product scoring against IVF-PQ ANN retrieval for this dual-encoder scoring function.
>
> What we will revise: In the paper we will (a) explicitly decompose inference cost into attention vs output-layer terms and state that our target is the O(N·d) softmax over items, and (b) add a short discussion of existing efficiency techniques such as sampled/approximate softmax, smaller distilled models, and candidate-generation plus re-ranking. We will explain that these reduce effective N within an ID-softmax formulation, while our approach reformulates the model as a dual encoder so that ANN search is the natural serving mechanism, making our method complementary to these solutions.
>
> Cold-start: model limitations and evidence
>
> Reviewer: “Argues that classical transformers struggle with item cold-start … no evidence is provided that the proposed model alleviates it.”
>
> Our clarification: Item cold-start is, by definition, a lack of interaction history, but current ID-softmax transformers are structurally unable to handle many common item cold-start scenarios: they simply cannot score items whose IDs never appear in training, even when rich item features are available. Their output layer has no parameters for unseen IDs, so those items are effectively invisible at inference time. RetrievalFormer instead encodes items directly from their attributes in a feature-based item tower, so it can embed and score items that were never seen in training.
>
> Evidence we already have (and will clarify better):
>
> LOOC protocol: We split items into train/val/test and remove all interactions for test items from training. Under this protocol, ID-softmax models cannot produce scores for test-only IDs at all, while RetrievalFormer can via its feature-based item tower. LOOC is used as a capability diagnostic, not a head-to-head comparison.
>
> 100% cold-start production dataset: On an email-campaign dataset where all recommended items are new at serving time, RetrievalFormer outperforms a strong content-based baseline using the same features, showing a practical cold-item benefit.
>
> Feature-only MovieLens variant: In a diagnostic experiment we train a version that removes user_id and item_id and uses only item/content features (including text) on MovieLens; it still achieves strong Recall@20, indicating the architecture remains competitive even when relying purely on content.
>
> What we will revise: We will adjust the abstract and contribution wording to state clearly that we demonstrate a feature-based dual-encoder design that can score unseen items and performs well in feature-rich cold-item settings, and explicitly note that standard ID-softmax transformers do not support such item cold-start scenarios. We will also mark ID-based baselines as “N/A (cannot score unseen IDs)” in the LOOC table and explain this point in the text.

---

> ### Author Response · Authors · 2025-12-01
> **Response to Reviewer r8bb - 2**
>
> Reviewer: “Implementation details … do not substantiate claims; there is no evidence demonstrating that the approach effectively tackles efficiency or cold-start issues.”
>
> Our clarification: The current presentation underplays the scope of the experiments. We evaluate RetrievalFormer against 12 different models across 3 public datasets (Amazon Beauty, Amazon Toys & Games, MovieLens-1M), including AttrFormer (a recent KDD 2025 model), and standard baselines such as GRU4Rec, SASRec, BERT4Rec, and others. We also report results on a fourth, real-world production dataset (email campaigns). Across these, RetrievalFormer achieves 86–91% of strong transformer baselines’ Recall@20 while being ANN-servable. During the rebuttal period we re-ran MovieLens-1M with the same setup and obtained an improved RetrievalFormer Recall@20 of 0.355 (vs 0.3022); we will update the table. The paper also contains several ablations (feature fusion vs simple pooling, shared vs separate embeddings, sequence length, batch size and negative sampling), plus the new feature-only variant mentioned above.
>
> What we will revise: We will expand the implementation section to clearly describe the training and serving configuration for the efficiency and cold-start experiments, including hardware, batch sizes, sequence lengths, negative sampling, and the exact FAISS/IVF-PQ parameters (number of lists, code size, nprobe, training size). We will also slightly reorganize Sec. 4 so that main benchmark results, the production experiment, and ablations are clearly separated, making the evidence easier to follow.
>
> Reviewer: “Paper appears hastily prepared and suffers from multiple technical and presentation shortcomings … ‘ANN’ not standard, question marks, layout issues, ambiguous ‘transformer accuracy’.”
>
> Our clarification: We regret these presentation issues create a poor impression. They are mostly formatting and citation artifacts, not missing content, and we will fix them. Specifically, we will (a) expand “ANN” to “Approximate Nearest Neighbor” at first mention and adjust the title/abstract wording, (b) remove stray “(?)” and “?” markers by restoring missing citations in the uniformity discussion and elsewhere, (c) reformat the table that exceeds layout bounds, (d) replace “transformer accuracy” with “accuracy of state-of-the-art transformer-based sequential baselines,” and (e) standardize terminology (e.g., consistently “MovieLens-1M”), remove odd characters such as the stray “L,” and thoroughly proofread the manuscript.
>
> Summary
>
> In summary, we clarify that (1) the computational bottleneck we target is the softmax over N items and we address it with a dual-encoder + ANN formulation that is complementary to existing efficiency techniques; (2) current ID-softmax transformers fundamentally do not support item cold-start, whereas our feature-based dual encoder can score unseen items, supported by LOOC, a 100% cold-start production dataset, and a feature-only variant; (3) the experimental study is broad and modern (12 models, 3 public datasets plus a production dataset, including a 2025 KDD model), and we will make implementation details fully explicit; and (4) the presentation issues that made the draft look rushed are specific and fixable, and we have concrete plans to correct them.

---

### Official Review · Reviewer_naFe · 2025-10-30

**Soundness:** 2
**Presentation:** 2
**Contribution:** 2
**Rating:** 2
**Confidence:** 4

**Summary:**

The paper proposed a dual encoder architecture for sequential recommendation that learns user action sequence embedding and item embedding separately and explicitly handles the recommendation as similarity search via dot product, hence enabling the use of vector search tools such as IVF and PQ for fast and large-scale search. The experiment highlights the query efficiency with near precision/recall. The overall proposed method makes sense on an intuitive level, although some claims are not well supported. Presentation, especially equations, can be improved.

**Strengths:**

1. Although not new, the authors studied a practically useful problem that moves beyond a static recommender system, where the sequential user interaction isn't utilized. The topic itself is impactful and is of broader interest to the machine learning community.

2. The idea is presented clearly, first formulating a recommendation as a similarity search between embeddings and logically bringing vector search tools such as IVF and PQ. This is a practically useful insight despite its engineering nature.

3. Evaluations are completed with many ablation studies, including feature fusion, LOOC, and others.

4. Most of the designs are well motivated and make sense on an intuitive level, despite their simplicity. To my best knowledge, bringing IVF-PQ style vector search in a sequential recommendation system is new and novel.

**Weaknesses:**

Major concerns:

1. It seems most of the gains claimed in the paper are from building an IVF+PQ structure for ANN search, which is not novel and is a well-known approach for recommender systems. The novelty of using this technique in a sequential recommendation system is very limited, and how IVF-PQ is incorporated into the proposed system is somewhat trivial.

2. One of my major concerns comes from the evaluation fairness: I found it unconvincing to state that previous works, such as SASRe, can not benefit from approximate vector search techniques; if this is the case, it needs to be examined more closely.

3. Recall/precision is results are not optimal; the author claims the main benefit is serving efficiency, but that claim is not well supported due to major concern#2. Also, the serving efficiency experiments feel underwhelming, as only one pretty old baseline is considered.

Other concerns:

1. Having separate embedding for items and users is not new in the recommendation systems, for sequential or non-sequential cases. Even for a fully transformer model, such as SASRec, this is done implicitly.

2. Alignment and uniformity statements regarding InfoNCE are strange and feel out of place. It is unclear what it means and what it brings to the paper. How uniformity brings good results is not clear and not well supported.

3. The data flow and model design figure is hard to understand. What exactly is AttentionFusion?
IVF-PQ details not given, how many inverted lists are there, and what is the configuration for the product quantization?

Small issues:

1. Citation issue: (?) appear in citation

2. “A fundamental challenge….” paragraph duplicated.

3. The LOOC term is used before defined.

4. Overall, the paper feels rushed and is in need of more polishing.

**Questions:**

1. Can Authors isolate and do an ablation study on the effect of IVF+PQ?

2. Why vector search techniques can’t be used in baselines?

---

> ### Author Response · Authors · 2025-12-01
> **Response to Reviewer naFe**
>
> We thank the reviewer for the thorough and constructive review and for highlighting the practical importance, clear formulation, and ablations. We address the key concerns below:
>
> 1. Dual‑encoder formulation vs using ANN with a transformer
> The serving regime we target is inherently retrieval‑style: we want to retrieve top‑K items via nearest‑neighbor search in a user–item embedding space, scale to large catalogs, and score cold items from features. Architecturally, this is the domain of dual‑encoder models, where a user encoder and an item encoder produce embeddings in a shared space.
>
> Standard sequential transformers such as SASRec or BERT4Rec are typically trained as ID‑softmax classifiers over a fixed item vocabulary: the output head is optimized to predict the next ID, not to form a retrieval‑ready embedding space. Their item representations, if extractable, and output layer are tightly bound to this ID softmax and anisotropic in nature.
>
> We agree that ANN is a general tool: any model with per‑item embeddings can, in principle, index them. However, simply attaching ANN to a softmax‑trained transformer does not yield a baseline with the same capabilities as RetrievalFormer
>
> It remains tied to a fixed item vocabulary. Indexing the existing ID embeddings does not create representations for completely unseen IDs, so such a model cannot score items in our LOOC protocol or in the 100% cold‑start production setting, regardless of exhaustive vs ANN scoring. RetrievalFormer instead has an item‑side encoder that consumes features and exposes a stable item embedding space, allowing it to encode unseen items from their attributes
>
> The training objective is not retrieval‑oriented. A softmax‑trained transformer optimizes classification, not retrieval quality in a shared user–item feature space. To obtain a transformer baseline with the same retrieval‑native, feature‑based properties, one would effectively need to re‑architect it into a dual encoder: add an item tower that consumes features rather than IDs, decouple item embeddings from the classifier head, and retrain user and item towers with a contrastive loss
>
> RetrievalFormer is exactly such a dual‑encoder adaptation: a transformer user tower + feature‑based item tower, shared embedding tables across feature contexts, and joint contrastive training so that the outputs are directly usable for ANN retrieval and cold items. We will make this distinction explicit in the paper: ANN itself is not unique to our method, but a transformer‑style sequential recommender explicitly cast and trained as a dual encoder for retrieval is
>
> 2. IVF‑PQ details and efficiency comparison
> We agree that IVF‑PQ is a standard ANN index and do not claim it as a contribution. Our latency study in Sec. 4.5 compares exhaustive dot‑product scoring vs IVF‑PQ retrieval for a dual‑encoder scoring function as the catalog size grows. The 288× speedup at 10M items reflects the structural difference between softmax over all items and ANN search, and would apply to other dual‑encoder models as well. We will add the missing FAISS configuration and clarify this framing in the text
>
> 3. LOOC and cold‑start evaluation
> LOOC is a strict, illustrative protocol: items are split into train/val/test, and all interactions for test items are removed from training, making them completely unseen. ID‑softmax transformers cannot score such items without being structurally modified into dual encoders (as above), so their Recall@k is undefined in this setting. Our goal with LOOC is to demonstrate that a feature‑based dual encoder can handle such items and to report its Recall@20, not to claim a definitive cold‑start benchmark. We will make this role explicit, label ID‑based baselines as “N/A (cannot score unseen IDs)” in the LOOC table, and de‑emphasize LOOC in the main text. We complement this with a 100% cold‑start production experiment against a strong content‑based baseline, where RetrievalFormer shows a sizable AUC improvement
>
> 4. InfoNCE, alignment/uniformity, and collapse
> We apologize for the confusing “uniformity (?)” markers. These are missing citations in Sec. 3.5 and Appendix C, not intentional notation. Appendix C defines alignment/uniformity and discusses representation collapse and the role of mixed negative sampling and batch size in dual encoders, summarizing known results. We will restore the citations, remove the "(?)" markers, and clearly state that this section summarizes known properties of InfoNCE that guide our design, rather than new theory
>
> 5. Diagram and minor issues
> We will revise the model diagram to clearly show the flow from raw features into AttentionFusion, then into the user/item towers and fix the duplicated paragraph, acronym usage, and citation formatting. These changes are presentational and do not affect the underlying results
>
> We hope these clarifications address your concerns about ANN usage, LOOC, and the InfoNCE discussion

---

### Official Review · Reviewer_ewbd · 2025-10-31

**Soundness:** 2
**Presentation:** 3
**Contribution:** 2
**Rating:** 2
**Confidence:** 4

**Summary:**

The paper presents RetrievalFormer, a two-tower recommender system combining a transformer-based user tower with a feature-based item tower to enable ANN-based retrieval. The item tower encodes heterogeneous item features via an attention fusion mechanism and shared embedding tables, making new item recommendation possible without retraining. The user tower models enriched interaction sequences with static profile features. The training uses InfoNCE with mixed negative sampling to learn a shared embedding space. Experiments show competitive accuracy relative to transformer baselines but with large efficiency gains (up to 288× speedup) and cold-start capability.

**Strengths:**

- Clear, thorough organization of the paper, with extensive architectural description and ablation studies.
- Experimental evaluation is broad (accuracy, cold-start, efficiency), includes comparisons against strong baselines, and covers real-world cases.

**Weaknesses:**

- The central “two-tower” design with shared embeddings is not new — similar architectures have been widely studied and deployed in the recommender systems literature.
- The attention fusion for heterogeneous features, while useful, is a minor variation on existing multi-head/self-attention feature interaction or set transformer designs, not a fundamentally new mechanism.
- The main claimed innovation is the framework combination rather than original algorithmic or modeling novelty, and the “efficient double-tower” approach is already common in related fields.
- As a result, the paper’s contribution lies mainly in engineering integration of known ideas rather than introducing new techniques.

**Questions:**

- How does RetrievalFormer’s attention fusion differ in measurable novelty from other feature interaction methods already used in two-tower models?
- Can the authors clarify in what sense the shared embedding design here is materially different from previous unified embedding table approaches?
- Would using standard two-tower architectures with similar feature fusion yield comparable results without the transformer component?

---

> ### Author Response · Authors · 2025-12-01
> **Response to Reviewer ewbd**
>
> We thank the reviewer for the clear summary and for noting the clarity of the architecture description and experimental breadth. We address the main concern about novelty and contribution scope.
>
> 1. Novelty vs known components.
> We agree that two‑tower architectures, shared embeddings, and attention‑based fusion are common components in recommender systems. Our contribution is not to introduce new primitives, but to reformulate sequential recommendation itself as a retrieval problem with a transformer‑based dual encoder and to show empirically that such a model can match the accuracy of strong transformer baselines while being naturally ANN‑servable and feature‑based. In RetrievalFormer:
>
> * the user tower is a transformer that performs full sequential modeling based on heterogeneous attributes;
> * the item tower encodes items from heterogeneous attributes via shared embeddings and AttentionFusion; and
> * both towers are trained jointly with a contrastive loss (used in retrieval tasks) so that recommendations are produced by user-item similarity in a shared embedding space.
>
> This means the same model is both the sequential recommender and the retrieval system. On Amazon and MovieLens‑1M, this retrieval‑native dual encoder attains similar Recall@20 to established transformer sequential models on the same data, while directly supporting ANN serving and feature‑based cold items.
>
> We will revise the introduction and contributions section to make this architectural reframing and empirical evidence the focal point and to explicitly state that we do not claim novelty for the underlying ANN or attention mechanisms.
>
> 2. Contribution vs “engineering integration.”
> We appreciate the concern that our work might be viewed as mostly integration. We see the main value in the insight that a retrieval‑first, transformer‑based dual encoder is a viable design point: it merges what is traditionally a separate “re‑ranking transformer” into the retrieval model itself, maintaining transformer‑level accuracy while enabling ANN‑based serving and feature‑based cold‑item scoring. We believe this work compliments the industry/domain as it proves that similar performance to SOTA methods can be achieved with a single architecture / system. We will sharpen this perspective in the paper so that the work is evaluated on this design‑level contribution and its empirical support, rather than on the novelty of individual components.

---

### Official Review · Reviewer_vn8j · 2025-10-31

**Soundness:** 2
**Presentation:** 2
**Contribution:** 2
**Rating:** 2
**Confidence:** 4

**Summary:**

This paper proposes RetrievalFormer, a dual-encoder architecture for sequential recommendation, aiming to combine the accuracy of Transformer-based models (e.g., SASRec, BERT4Rec) with the serving efficiency of approximate nearest neighbor (ANN) retrieval. The model consists of a Transformer-based user tower that encodes user interaction sequences, and a feature-based item tower that encodes heterogeneous item attributes. Both towers share embedding tables to maintain semantic consistency. The paper’s key novelty lies in an attention-based heterogeneous feature fusion mechanism that dynamically integrates diverse feature modalities (text, category, numeric, etc.) for both user and item representations.

**Strengths:**

(1) The paper addresses a practically important and relatively underexplored problem—improving inference efficiency in Transformer-based sequential recommenders—by integrating retrieval mechanisms into the modeling framework.
(2) The work provides both experimental and theoretical support, including ablation studies, detailed architecture analysis, and discussion of InfoNCE training dynamics, which lend credibility to the proposed approach.
(3) The paper is well-structured overall, with clear sectioning, comprehensive appendices, and reproducibility details that make it easier to follow the methodology and replicate the experiments.

**Weaknesses:**

1. Fragmented problem formulation and insufficiently articulated contributions.
The paper identifies two key challenges—low retrieval efficiency and item cold-start—but these are treated as independent objectives rather than as parts of a unified framework. The proposed dual-tower design and ANN-based retrieval address efficiency, while feature-based item encoding targets cold-start resilience. However, these components appear loosely coupled, lacking a cohesive theoretical or algorithmic integration.

2. Weak formalization and unclear writing.
(1) Unclear notation: Equations (5–7) introduce undefined and nonstandard terms such as AttentionFusion, ItemFeatures, and InteractionContext, which are not formally explained at their first occurrence.
(2) Formatting and typographical issues: e.g., Line 260 “enforces uniformity (?)”; Line 461 “while achieving ? 90%”; Line 326 “averaged over five runs with std. < 0.001 not reported.”
(3) Lack of rigor in description: e.g., Line 81 “while achieving 288× speedup at 10M items” does not specify the comparison baseline; Line 285 ambiguously uses “ML-1M” without clarifying it refers to MovieLens-1M.
(4) Unclear visualization: Figure 1 does not explain the color-coded “Categorical Feature” elements and fails to depict the central ANN retrieval process, a key claimed contribution.

3. Incomplete experimental validation.
(1) For RQ3 (Cold-start Item Recommendation), the analysis compares only the model’s own performance under LOO vs. LOOC settings, omitting classical cold-start baselines (e.g., content-based or attribute-aware methods).
(2) For RQ4 (Efficiency and Latency), efficiency is compared solely against SASRec, excluding stronger Transformer-based baselines such as AttrFormer and DIF-SR, which achieve higher accuracy (Table 1) and are more relevant for evaluating the claimed efficiency–accuracy trade-off.

4. Insufficient efficiency analysis.
The paper does not explicitly compare inference time under softmax-based scoring vs. ANN-based retrieval. Such analysis would clarify whether efficiency gains stem from the model design or simply from substituting softmax with ANN—a change that could, in principle, also accelerate conventional models like SASRec.

5. Unclear handling of heterogeneous feature types.
The paper claims support for diverse modalities (text, categorical, numerical, and interaction features) but only briefly mentions embedding features with “shared lookup tables” (Line 159). It remains ambiguous how these modalities are processed—e.g., whether textual features use pretrained embeddings, token averaging, or learned representations.

6. Missing comparison with traditional two-tower retrieval models.
While the Related Work section distinguishes RetrievalFormer from matrix-factorization-based dual encoders, no empirical comparison is provided with classical two-tower retrieval baselines. Such results are necessary to substantiate the claimed benefit of introducing Transformer-based sequence modeling into a retrieval-oriented framework.

**Questions:**

See Weaknesses

---

> ### Author Response · Authors · 2025-12-01
> **Response to Reviewer vn8j**
>
> We thank the reviewer for the careful and detailed review. We respond to the main points below.
>
> 1. Fragmented treatment of efficiency and cold‑start.
> We agree that, as currently written, the introduction presents efficiency (full softmax over all items) and cold‑start (new items) as if they were separate concerns. Conceptually, they share a common root: standard sequential transformers treat recommendation as classification over a fixed item‑ID vocabulary. This leads to (i) an O(N·d) output cost from scoring every item in the softmax head, and (ii) an inability to score items whose IDs never appear during training. RetrievalFormer addresses both by recasting sequential recommendation as dual‑encoder retrieval: a transformer user tower encodes the interaction history; a feature‑based item tower encodes item attributes; and a contrastive objective trains a shared user–item embedding space so that ANN search directly yields recommendations. This eliminates the full softmax and enables feature‑based scoring of unseen items within a single architecture. We will revise Sec. 1 to make this coupling explicit and present efficiency and cold‑start as two manifestations of the same underlying design choice.
>
> 2. Formalization and unclear writing .
> The "uniformity (?)" markers in Sec. 3.5 and Appendix C and the stray "?90%" are artifacts of missing citations and conversion issues, not intended notation. Appendix C already defines alignment/uniformity and representation collapse for InfoNCE‑trained dual encoders; in the revision we will restore the missing citations, remove the "(?)" markers, and clearly state that we are summarizing known properties used to justify design choices (e.g., mixed negative sampling), not proposing new theory. We will also (i) standardize dataset names (always "MovieLens‑1M"), (ii) clarify Table 1’s caption so it is clear that baseline figures are 5‑run averages from prior work while our RetrievalFormer numbers are single runs, and (iii) revise Fig. 1 to more clearly depict the ANN retrieval step and explain the color coding for different feature types.
>
> 3. Experimental validation: LOOC, efficiency, and missing baselines.
> For LOOC, we split items into train/val/test and remove all interactions involving test items from training. Under this protocol, ID‑softmax transformers (SASRec, BERT4Rec, etc.) cannot assign scores to test items: those IDs simply do not exist in their learned output vocabulary. Their Recall@k is therefore undefined (or trivially zero) unless we structurally modify them. LOOC is meant as a capability diagnostic: it illustrates that a feature‑based dual encoder like RetrievalFormer can, in principle, handle items that are completely unseen at training time, and it quantifies the resulting Recall@20 in that regime. We will make this intent explicit, label ID‑based baselines as "N/A (cannot score unseen IDs)" in the LOOC table, and move some details to the appendix so LOOC is not over‑emphasized as a central benchmark.
>
> For efficiency (Sec. 4.5), our comparison is between exhaustive dot‑product scoring and ANN retrieval as the catalog size grows. The measured 288x speedup at 10M items reflects the scaling difference between O(N) scoring and ANN search, and would apply broadly to any model that exposes a dual‑encoder form.
>
> We agree that including a classical two‑tower baseline using the same features would further contextualize the benefit of the transformer elements. In this submission we prioritized strong transformer baselines because they define the accuracy "ceiling" we aim to match; we will expand our discussion of how RetrievalFormer relates to traditional two‑tower models and treat adding such baselines as priority future work. We believe adding an additional vector here might further complicate the paper and may not be a scientifically rigorous comparison.
>
> 4. Heterogeneous feature handling and shared embeddings.
> We acknowledge that the main text does not fully surface this design as many details are pushed to appendix. Sec. 3.2 introduces the AttentionFusion module, which embeds each categorical feature via a feature‑specific table, projects all features into a common hidden dimension, and applies multi‑head self‑attention and pooling. Appendix B.2 further specifies that the same embedding tables are reused across user profile, item metadata, and interaction context, with appropriate aggregation for single‑ vs multi‑valued features. In the revision, we will move the essential parts of Appendix B.2 into Sec. 3.2 and add a small schematic to Fig. 1 to clearly show how heterogeneous features are handled and how embedding tables are shared across towers
>
> We hope these clarifications address your concerns about the integration of efficiency and cold‑start, formalization, and experimental coverage

---

### Author Response · Authors · 2025-12-01
**Author response summary: contributions**

We thank the reviewers for their detailed and thoughtful feedback. Below we summarize what our paper does, what we see as its main contributions, and how we address the central concerns raised in the reviews. Reviewer‑specific replies follow separately.

What the paper does (brief overview):

Modern large‑scale recommenders are often two‑stage systems: a simple retrieval model picks candidates; a heavier sequential transformer re‑ranks them. This implicitly assumes that transformer‑level sequence modeling and retrieval‑native serving cannot live in the same model.

Our paper asks a different question:

Can we treat sequential recommendation itself as a retrieval problem, using a transformer‑based dual encoder that is both ANN‑friendly and feature‑based, while achieving similar accuracy to strong transformer baselines?

We propose RetrievalFormer, a dual‑encoder architecture where:

A transformer user tower models the feature-based interaction sequence based on the heterogeneous features found in the items a user interacts with.

A feature‑based item tower encodes items from heterogeneous attributes (including content features) using shared embedding tables and an attention‑based fusion module.

The two towers are trained jointly with a retrieval‑oriented contrastive objective, so that user and item representations lie in a shared embedding space and recommendations are produced directly by approximate nearest neighbor (ANN) search rather than a softmax over item IDs.

This makes RetrievalFormer retrieval‑native, allows it to score completely unseen items from their features, and lets us serve predictions using standard ANN libraries.

Main contributions and findings

Reframing sequential recommendation as dual‑encoder retrieval.
Standard sequential transformers (e.g., SASRec, BERT4Rec) are trained as ID‑softmax classifiers over a fixed item vocabulary: the output head is optimized to predict the next ID, not to form a retrieval‑ready embedding space. In contrast, RetrievalFormer is explicitly cast as a dual‑encoder retrieval model: the user and item towers are symmetric, feature‑based, and trained with a contrastive loss so that ANN search over item embeddings is the recommendation mechanism. This is a different formulation of sequential recommendation: retrieval, not classification.

Transformer‑level accuracy in a retrieval‑native architecture.
On standard benchmarks (Amazon Beauty/Toys, MovieLens‑1M), RetrievalFormer achieves similar Recall@20 to strong transformer baselines on the same data while being directly ANN‑servable. In the original submission we reported ~86–91% of transformer Recall@20; during the response period we slightly improved our MovieLens‑1M configuration (same data and feature setup as in the paper), obtaining a RetrievalFormer Recall@20 of 0.355 instead of 0.3022. We will update the table accordingly. This does not change any conclusions but shows that the architecture is at least as strong as originally reported and can approach the “transformer cluster” more closely, while greatly exceeding the efficiency.

Unified retrieval‑focused recommender design.
Because RetrievalFormer is both transformer‑based and retrieval‑native, the distinction between “retriever” and “re‑ranker” blurs: the same dual encoder both models the user sequence and retrieves items via ANN. Our results show that such a model can approach the accuracy of transformer re‑rankers while being directly deployable as the retrieval engine, suggesting a path toward unified, retrieval‑focused recommenders in regimes where latency and cold‑items matter, rather than strictly two‑stage pipelines.

Feature‑based item scoring and cold‑item capability.
Items are encoded from their content and attribute features, not bound to a fixed ID vocabulary. This allows RetrievalFormer to score items that are completely unseen during training, both in a strict LOOC protocol (where test items and their interactions are removed from training) and in a 100% cold‑start production dataset (email‑campaign recommendations) where all recommended items are new at serving time. RetrievalFormer outperforms a strong content‑based baseline in this production setting.
During the response period we also ran a diagnostic feature‑only variant on MovieLens‑1M that removes user_id and item_id and uses only item/content features (including text). This configuration achieves Recall@20 = 0.398, showing that the architecture can remain highly competitive even when it must rely purely on content rather than IDs, further supporting the feature‑based, cold‑item story.

---

### Meta-Review · Area_Chair_45Px · 2026-01-07

**Summary:**

The reviewers were unanimous in their initial skepticism, centering on three key issues: limited novelty, presentation quality, and baseline fairness.
- Novelty: Multiple reviewers (ewbd, naFe) felt the work was an engineering combination of known components (Two-Tower, Transformers, ANN) rather than a fundamental algorithmic advance.
- Presentation: The draft suffered from significant polish issues, including missing citations (rendering as "?"), undefined notation, and unclear diagrams, which hurt the paper's credibility (vn8j, r8bb).
- Baselines: Reviewers questioned why standard Transformers (like SASRec) couldn't simply use ANN, implying the efficiency comparison was unfair.

**Reviewer Concerns:**

Addressed by Rebuttal
- The "SASRec + ANN" Misconception: The authors effectively clarified that standard sequential transformers are trained as ID-classifiers (anisotropic space), meaning one cannot simply plug in ANN for retrieval without retraining as a dual-encoder—which is exactly what RetrievalFormer does.
- Cold-Start Definition: The authors successfully countered r8bb’s claim that cold-start is "only a data issue," explaining the structural limitation of ID-based models (which have no parameters for unseen IDs) versus their feature-based approach.
- Efficiency Metrics: They clarified that the 288x speedup is a comparison against exhaustive dot-product scoring, validating the utility of the dual-encoder formulation.

Outstanding
- Fundamental Novelty: While the authors argue their contribution is the reframing of sequential recommendation as a retrieval problem, the reviewers' view that this is "engineering integration" rather than ICLR-level research likely remains unchanged.
- Presentation Trust: While the authors promised to fix the formatting and citation errors, the sheer volume of "hasty" errors noted by r8bb and vn8j makes it difficult to trust the rigor of the submission without seeing a revised PDF.

**Reviewer Scores:**

Reviewer vn8j: 2 → 4

Reviewer ewbd: 2 → 4

Reviewer naFe: 2 → 4

Reviewer r8bb: 2 → 2

---

### Decision · Program_Chairs · 2026-01-26

Reject